# Invasive alien pests threaten the carbon stored in Europe's forests

Rupert Seidl [1], Günther Klonner[2], Werner Rammer[1], Franz Essl[2], Adam Moreno[1,3], Mathias Neumann [1] & Stefan Dullinger[2]

Forests mitigate climate change by sequestering large amounts of carbon (C). However, forest C storage is not permanent, and large pulses of tree mortality can thwart climate mitigation efforts. Forest pests are increasingly redistributed around the globe. Yet, the potential future impact of invasive alien pests on the forest C cycle remains uncertain. Here we show that large parts of Europe could be invaded by five detrimental alien pests already under current climate. Climate change increases the potential range of alien pests particularly in Northern and Eastern Europe. We estimate the live C at risk from a potential future invasion as 1027 Tg C (10% of the European total), with a C recovery time of 34 years. We show that the impact of introduced pests could be as severe as the current natural disturbance regime in Europe, calling for increased efforts to halt the introduction and spread of invasive alien species.

[1] Institute of Silviculture, University of Natural Resources and Life Sciences (BOKU) Vienna, Peter Jordan Straße 82, Wien 1190, Austria. [2] Department of Botany and Biodiversity Research, Faculty of Life Sciences, University of Vienna, Rennweg 14, Vienna 1030, Austria. [3] NASA Ames Research Center, Mail Stop 204-14, Moffett Field, CA 94035-0001, USA. Correspondence and requests for materials should be addressed to R.S. (email: rupert.seidl@boku.ac.at)

Terrestrial ecosystems regulate the climate via uptake and storage of carbon (C) from the atmosphere[1,2]. C sequestration of forest ecosystems has compensated 60% of anthropogenic C emissions between 1990 and 2007[3]. Currently, the total C stored in forest ecosystems globally exceeds the C in the atmosphere[3]. Although these forest C stores are long-lived, they are not permanent[4]: disturbances (i.e., pulses of tree mortality) can lead to a rapid and substantial release of C back to the atmosphere[5,6]. Disturbances can cause direct C emissions from biomass combustion in wildfires[7]. However, forest ecosystems also loose C in the wake of disturbances from insects and diseases, as a result of increased heterotrophic respiration[8] and decreased C uptake[9]. As disturbances are highly climate sensitive[10], such a disturbance-mediated C loss could result in amplifying climate feedbacks, i.e., C release from disturbances further fueling climate change, which in turn increases forest disturbance activity[11,12]. Understanding the interactions between the climate system and forest disturbance regimes is thus of paramount importance for quantifying the potential future contribution of forests to climate change mitigation.

In recent decades, transcontinental human trade has removed many dispersal barriers for species[13] and has led to a global redistribution of forest pests[14]. Alien pest species can cause particularly severe tree mortality because they often lack natural enemies in their new range and meet naive hosts that have not adapted to these pests through coevolution[15]. In extreme cases, alien pests can virtually eliminate tree species from their entire geographical range (e.g., chestnut blight[16] in North America). Climate change is expected to further aggravate the severity of invasive alien pests, as warmer temperatures and increased tree stress (e.g., due to drought) likely facilitate their establishment and spread[17,18]. It is thus important to not only account for the effect of natural disturbances on climate regulation[12] but also consider the potential climate-mediated emergence of novel disturbance regimes, consisting of newly introduced agents of tree mortality. However, the compound effects of alien pest invasions and climate change remain poorly quantified. This limits our ability to address invasive alien pests in forest management in order to safeguard the climate regulating function of forests.

Here we project the potential distribution of five invasive alien pest species in Europe's forests under current and potential future climate conditions, and estimate the C cycle consequences of these novel disturbance agents. We focus on five of the most detrimental invasive pest species for Europe's forests, all of which have widely distributed and economically important tree species as their hosts. The selected pests are alien to Europe but have already established populations in limited areas of the continent. They include species from different taxonomic groups, i.e., an insect (Asian Longhorned Beetle, ALB), a nematode (Pine Wood Nematode, PWN), two oomycetes (Sudden Oak Death, SOD and Beech Bleeding Canker, BBC), and a fungus (Pitch Pine Canker, PPC). We modeled the potential distribution of these pests in Europe under current climate (1950–2000) and future scenarios of climate change (2030–2080) (Supplementary Table 1), using species–climate relationships from occurrence data collected in both the native and alien ranges of the respective pests. Subsequently, we combined the areas of potential pest occurrence with spatial data on the distribution of live tree C in Europe's forests[19] to assess potential continental-scale C cycle impacts based on three indicators: First, the potential live tree C at risk from a complete invasion of the regions climatically suitable for each pest was derived from combining projected pest distribution maps with maps of host tree C distribution and accounting for pest-specific mortality rates. Live tree C at risk thus gives an upper bound of the potential ecosystem impacts of alien pests. Second, for evaluating the consequences of novel disturbances it is essential to also account for the ability of ecosystems to recover from these potential impacts. In the context of C cycle effects, the time needed to recover the potential live C loss via net primary production (NPP) is a key attribute. Here we used remotely sensed net primary productivity in combination with C cycle modeling[20] to determine C recovery times. Third, we estimated the equilibrium C cycle effect of novel disturbance regimes of alien pests, i.e., to what degree alien pests reduce forest C storage relative to undisturbed conditions, taking into account disturbance severity, the fate of the disturbed C, as well as the potential of the forest to recover through C uptake. While the first two analysis steps focused on live tree C as response variable (i.e., an indicator for the ecosystem impacts of alien pests), this final step addressed disturbance effects on total ecosystem C (i.e., an indicator of relevance in the context of the climate regulating function of forest ecosystems). The equilibrium C cycle effect of invasive alien pests assessed in step three was compared to the C cycle impacts of the natural disturbance regime from wind, native bark beetles, and wildfire in Europe's forests.

## Results

**Potential distribution of alien pests**. We found that, under current climatic conditions, almost the entire European continent could be invaded by at least one of the five pest species studied (Table 1). Many areas of Western and Central Europe are climatically suitable for several alien pests (Fig. 1). At present, ALB has the largest potential distribution in Europe (3.17 Mill. km$^2$), followed by PPC and PWN, all of which could invade areas >1 Mill. km$^2$ (Supplementary Table 2). SOD and BBC are climatically restricted to Western Europe under current climate. An intermediate warming scenario (representative concentration pathway (RCP) 4.5, average temperate change +1.4 °C in 2030–2080) increases the climatically suitable area for all five pest species. The pine-dwelling species PWN and PPC respond most strongly to climate change and expand their potential range by +55.3% and +49.7%, respectively, while ALB—with a wide potential range already under current climate—changes only marginally (+2.7%). The direction of the projected range expansion is primarily to the east and north for all five species (Fig. 1). Under severe climate change (RCP 8.5, +2.4 °C), the climatically suitable areas increase further for all pest species except ALB, and even a moderate climate change scenario (RCP 2.6, +1.1 °C) still resulted in increasing potential pest habitat relative to current climate (Supplementary Table 2).

**Live tree C at risk**. ALB and PWN were the most important alien pest species with regard to live tree C at risk under both current and future climate conditions. We project that 596 Tg C (i.e., 26.8% of the live tree C currently stored in the two-needle pine forests of Europe) are at risk from an invasion of PWN, and another 376 Tg C are at risk from ALB (scenario RCP 4.5). Together, these two species account for 94.7% of the live tree C at risk from all five forest pests taken together, which amounts to 1027 Tg C (i.e., 10.4% of the C currently stored in Europe's live tree biomass) under the intermediate warming projected by the RCP 4.5 scenario (Fig. 1, Supplementary Fig. 1). One third of this live tree C at risk can be attributed to the effect of extended climatically suitable areas of pest species under climate change (Supplementary Table 3). Considering the uncertainties arising from within-grid cell variation in live tree C stocks and pest-specific variation in mortality rates, the 95% uncertainty interval of live tree C at risk ranged from 955 to 1102 Tg C (i.e., varying by −7.0% to +7.3%). Relative uncertainty increased with decreasing C at risk and was highest for SOD (Table 2).

**Table 1 Characteristics of the studied invasive forest pests**

| Species | Organism and symptoms | Origin and biology | Potential hosts in Europe | Mortality | Management |
|---|---|---|---|---|---|
| ALB Asian Long-horned Beetle (*Anoplophora glabripennis*) |  | Beetle native to temperate forests of Eastern Asia, larval develop in trunks and branches of trees, eventually leading to tree death | *Acer* ssp., *Aesculus hippocastanum, Betula* spp, *Platanus* spp., *Populus x canadensis, Populus nigra* | 80% (70-90%) | Identification of infested trees using sniffer dogs, removal of infested trees, quarantining infested areas |
| PWN Pine Wood Nematode (*Bursaphelenchus xylophilus*) |  | Nematode native to North America, adults live and reproduce in the wood of infected trees, eventually disrupting water transport | *Pinus nigra, P. pinaster, P. sylvestris* | 85% (80-90%) | Quarantining infested areas, removal of infested trees, controlling the vector species (*Monochamus* ssp.) |
| SOD Sudden Oak Death (*Phytophthora ramorum*) |  | Oomycete with unknown native range, causing stem cankers, leaf blight and branch dieback | *Betula pendula, Fagus sylvatica, Larix decidua, Pseudotsuga menziesii* | 5.5% (1-10%) | Removal of infested trees, quarantining infested areas to limit spread |
| BBC Beech Bleeding Canker (*Phytophthora kernoviae*) |  | Oomycete with unknown native range infecting the phloem of plants, leading to bleeding lesions, bark necrosis and cancers | *Fagus sylvatica, Quercus ilex, Q. robur* | 3% (1-5%) | Removal of infested plants, quarantining infested areas, controlling the main host species (*Rhododendron* ssp.) |
| PPC Pitch Pine Canker (*Fusarium circinatum*) |  | Fungus probably native to North America, infecting roots and branches, causing yellowing of needles, shoot dieback and cankers | *Pinus halepensis, P. nigra, P. pinaster, P. sylvestris* | 3% (1-5%) | Removal of infested trees, quarantining infested areas to limit spread |

Information derived from refs. [21,22]; mortality relates to unmanaged infestations
ALB (left) sourced from https://www.flickr.com/photos/99758165@N06/14770405395; released under a Creative Commons Attribution 2.0 Generic license; ALB (right) sourced from Larry R. Barber, USDA Forest Service, Bugwood.org; released under a Creative Commons Attribution 3.0 license; PWN (left) sourced from https://commons.wikimedia.org/wiki/File: Bursaphelenchus_xylophilus_male_tail.jpg; released under a Creative Commons Attribution 3.0 Unported license; PWN (right) sourced from USDA Forest Service—North Central Research Station, USDA Forest Service, Bugwood.org; released under a Creative Commons Attribution 3.0 license; SOD (left) and BBC (left) sourced from Widmer, T. L. 2010. Differentiating *Phytophthora ramorum* and *P. kernoviae* from other species isolated from foliage of rhododendrons. Online. Plant Health Progress https://doi.org/10.1094/PHP-2010-0317-01-RS. Available in the public domain; SOD (right) sourced from Joseph OBrien, USDA Forest Service, Bugwood.org; released under a Creative Commons Attribution 3.0 license; BBC (right) sourced from https://commons.wikimedia.org/wiki/File: Phytophthora_kernoviae_-_Beech_tree_infection.jpg?uselang=de, Forestry Commission; released under the Open Government License; PPC (left) sourced from https://commons.wikimedia.org/wiki/ File:Esporas_de_Fusarium_circinatum.png; released under a Creative Commons Attribution Share-Alike 4.0 license; PPC (right) sourced from Terry S. Price, Georgia Forestry Commission, Bugwood.org; released under a Creative Commons Attribution 3.0 license

**C recovery and pest management**. We found C recovery times to range from <1 year up to several decades for the five pest species, with the longest recovery times for ALB and PWN (Supplementary Table 4 and Supplementary Fig. 2). Considering the effect of all five pest species together, the average recovery time under a scenario of intermediate climate change is 34 years. Recovery is slowest in boreal forests (Fig. 2), which are relatively species poor and have lower annual C uptake than temperate ecosystems. Forests in the Mediterranean biome are able to recover most rapidly, due to smaller C stocks, shorter C residence times, and a lower impact of the five pest species in southern Europe.

Once pest species have been introduced, targeted management measures are crucial to reduce their impact. To quantify the potential effect of pest management we harnessed previous experiences in managing the studied pest species in invaded areas (Supplementary Table 5). Considering all five species jointly, we found that pest management can decrease the live tree C at risk to approximately 1/4 of the unmanaged scenario (i.e., 170 Tg C and 257 Tg C under current climate and intermediate climate change, respectively) (Fig. 3, Supplementary Table 3). This reduces C recovery times from 34 to 19 years under intermediate climate change (Supplementary Table 4, Supplementary Fig. 2).

**Comparison to the impacts of natural disturbances**. Equilibrium C cycle effects—accounting for pest impacts as well as the fate of the affected tree C and forest recovery—were considerably below the upper bound estimates for live tree C at risk. However, the ranking of alien species impacts was consistent, with ALB and PWN exerting the largest reduction to the equilibrium C storage capacity of forests. Overall, a complete invasion by all five pest species would reduce the long-term C storage potential of

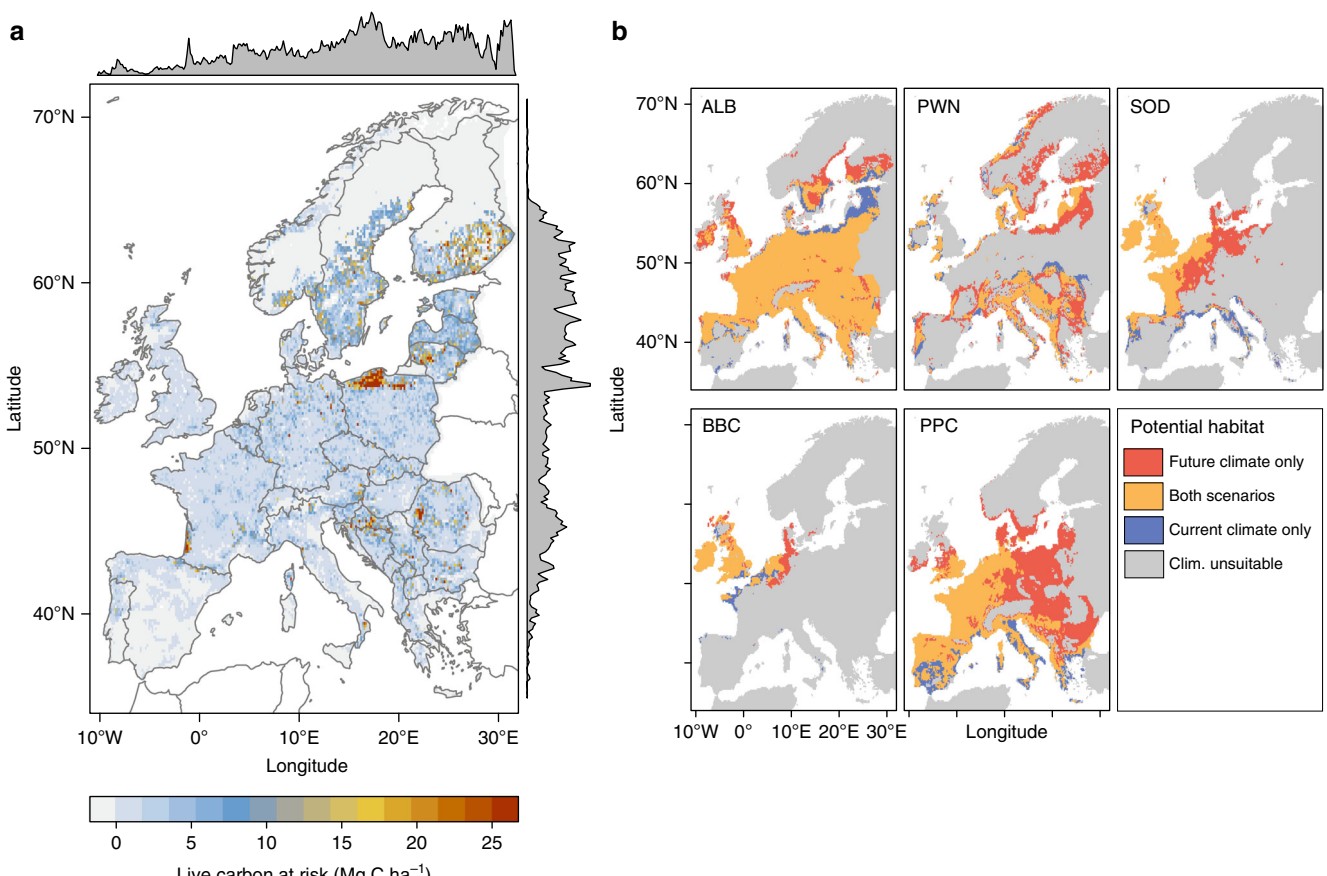

**Fig. 1** Live tree carbon at risk from an invasion of five alien pest species into their climatically suitable areas in Europe. **a** The total amount of live tree carbon at risk (in Megagrams carbon per hectare) from a complete invasion of all five pest species into their climatically suitable areas under intermediate climate change (2030–2080, scenario RCP 4.5). **b** Climatically suitable ranges for each pest species under current climate (1950–2000) and intermediate climate change (2030–2080, scenario RCP 4.5). ALB: Asian Long-horned Beetle (*Anoplophora glabripennis*), PWN: Pine Wood Nematode (*Bursaphelenchus xylophilus*), SOD: Sudden Oak Death (*Phytophthora ramorum*), BBC: Beech Bleeding Canker (*Phytophthora kernoviae*), PPC: Pitch Pine Cancer (*Fusarium circinatum*)

Europe's forests by 309 Tg C and 393 Tg C under current and future (RCP 4.5) climate conditions, respectively, when pest management measures are considered (see Supplementary Table 6 for unmitigated effects). The potential impact of invasive alien pests is thus in the same order of magnitude as the one estimated for the current natural disturbance regime in Europe's forests (Table 3).

## Discussion

Large parts of Europe are conducive to an invasion by alien pest species already under current climate. This finding is well in line with growing reports of alien pests establishing themselves on the continent[21,22] and the observation that the spread of alien species continues to increase[23]. It furthermore underlines that anthropogenic agency is the primary cause of the increasing global proliferation of species and that managing introductions is of paramount importance[13]. Here we show that climate change further aggravates the issue by relaxing climatic habitat limitations and considerably extending the potential distribution of important alien pest species[24,25]. Even areas where the alien pests considered here were of limited relevance in the past, such as in Northern and Eastern Europe, could be increasingly susceptible to invasion as a result of climate change[14].

Uncertainties remain regarding the future distribution of the modeled pest species. As currently available data on species

occurrence (cf. Supplementary Figs 3–7) frequently lack information on the specific reference period of observation, we here were able to only consider long-term averages of climatic conditions in the modeling of potential alien species distribution. While these climatic averages are indicative of the potential range of a species, outbreaks of pests are frequently associated with climatic variation, such as drought periods[26,27]. Future work should thus also consider the effect of climate variation in order to increase the robustness of projected invasive pest dynamics under climate change. Furthermore, the correlative modeling approach used here disregards spatio-temporal population dynamics, i.e., the processes of demography and dispersal, which limit the rate of spread. Approaches that include such dynamics exist but require extensive parameterization data that are currently available only for a small set of pest species and restricted locations (e.g., SOD in California[28]). While our results highlight the potential magnitude of alien pest effects on forest ecosystems, they hence do not provide forecasts of mid-century pest species distributions in Europe. To assess C cycle impacts of invasive alien pests at the continental scale, we here used a simple theoretical model[20]. While such an approach is not able to fully capture the intricacies of the forest C cycle[12,29], it is relatively robust to the considerable input data uncertainty at large spatial scales and facilitates a comparison of the relative effects of different forcings (e.g., disturbance agents, management, climate) on forest C storage.

It is important to note that our modeling approach does not account for possible interactions within pest populations (e.g., density-dependent effects) or with pathogens and parasitoids of the pest species under novel climates[30]. Consequently, potentially increasing pest impacts under increasing pest density and diversity are not considered[31]. In turn, also possible dampening feedbacks from antagonists are neglected. Such dampening feedbacks are particularly difficult to anticipate as they are not necessarily restricted to currently present antagonists but could also result from the regional emergence of new antagonists that are invaders themselves[32]. Finally, the occurrence data used to calibrate distribution models originate to a large degree from the species' invaded ranges rather than from their native ranges, as the ecological effects of pests are frequently larger[15] and species thus more likely to be observed in their invaded range. It remains unclear, however, to which degree pest species have already reached a distributional equilibrium with climate in these invaded ranges. A possible disequilibrium would result in models that do

not capture the species' full climatic niches and hence in projections that underestimate their potential ranges[33]. While our results are likely to overestimate actual pest species distributions in 2050 because we disregard dispersal limitations[34], they—resulting from the bias in the underlying observations—are conservative with regard to the potential climatically suitable ranges that these species could occupy in Europe. Further contributing to a conservative potential range estimate is the utilization of consensus predictions (rather than a maximum set) from different modeling algorithms (cf. Supplementary Fig. 8).

Europe's forests have mitigated climate change by sequestering substantial amounts of C from the atmosphere in past decades[35,36]. More recently, Europe's forest C sink has weakened, partly as a result of increased disturbances[6,37]. We found that, in the worst-case scenario of an unmitigated invasion of their entire climatically suitable area, a single aggressive pest species such as ALB or PWN can have C cycle impacts that are in the same order of magnitude as those of the three currently most important natural disturbance agents in Europe's forests—fire, wind, and native bark beetles—combined. As climatically suitable ranges increase under climate change, such invasions could cause substantial amplifying feedbacks to climate change through changes in forest demographics[4] and C loss from the biosphere[5,11].

The potential interactions between natural and alien agents of tree death remain poorly understood[38]. For example, whether alien pest species will increase the impact of other disturbance agents (e.g., by weakening trees and increasing their susceptibility) or reduce them (e.g., by competing for the same resources) remains unclear. Furthermore, uncertainties remain regarding which tree species will fill the niches opened by novel disturbances, and how the emerging dynamics will change Europe's forest composition[39]. Also climate change is likely to alter the future composition, structure, and C density of forests[40,41], yet we here assessed the potential C at risk based on the current state of Europe's forests (see Supplementary Table 7 for a sensitivity analysis). Furthermore, future changes in climatic extremes[42] could also affect the distribution of and susceptibility to alien pest species but are not considered explicitly here. An improved understanding of the interactions between climate, forest vegetation, and (novel as well as native) disturbance agents is thus

**Table 2 Live tree C at risk from invasive alien pest species and its uncertainty**

| Species | Current climate | | Future climate | |
|---|---|---|---|---|
| | Live tree C at risk | Uncertainty range | Live tree C at risk | Uncertainty range |
| ALB | 387.4 | 342.2–433.8 | 376.4 | 331.7–420.4 |
| PWN | 280.9 | 264.8–297.0 | 596.2 | 562.6–629.4 |
| SOD | 8.2 | 1.8–14.7 | 24.5 | 5.4–43.5 |
| BBC | 2.8 | 1.1–4.6 | 4.6 | 1.7–7.6 |
| PPC | 11.6 | 4.3–18.8 | 38.6 | 14.1–63.0 |
| All | 686.4 | 634.6–741.8 | 1026.9 | 955.1–1102.4 |

Uncertainties considered are the local (i.e., within-grid cell) variation in live tree C stocks as well as variation in pest-specific mortality rates. Uncertainties are calculated via bootstrapping, and the 2.5th to 97.5th percentile range is reported. Current climate refers to the period 1950–2000, and future climate to the period 2030–2080 under an RCP 4.5 scenario. Values disregard the effect of pest management and represent unmitigated live tree C impacts in Tg C
ALB: Asian Long-horned Beetle (*Anoplophora glabripennis*), PWN: Pine Wood Nematode (*Bursaphelenchus xylophilus*), SOD: Sudden Oak Death (*Phytophthora ramorum*), BBC: Beech Bleeding Canker (*Phytophthora kernoviae*), PPC: Pitch Pine Cancer (*Fusarium circinatum*), All: upper bound of live tree C at risk from all five invasive alien pest species jointly

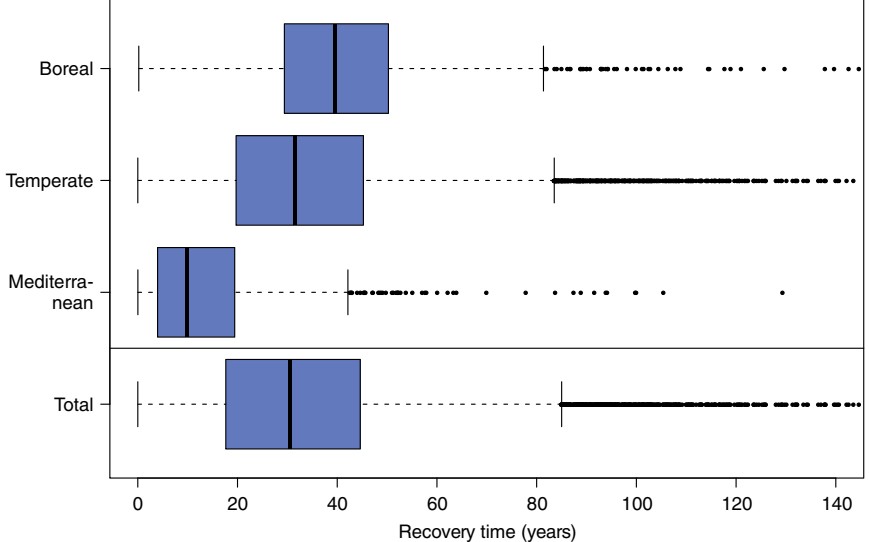

**Fig. 2** Time needed to recover the potential live tree carbon loss from an invasion of five alien pest species in Europe's forests. Values assume that pest species invade all areas suitable under an intermediate climate change scenario (RCP 4.5). Impacts of pest species are assumed to be unmitigated by management. Values of >150 years were truncated for clarity. Total refers to the entire European continent. Boxes show the median and interquartile range of the data; whiskers extend to the most extreme data point which is no more than 1.5 times the interquartile range from the box

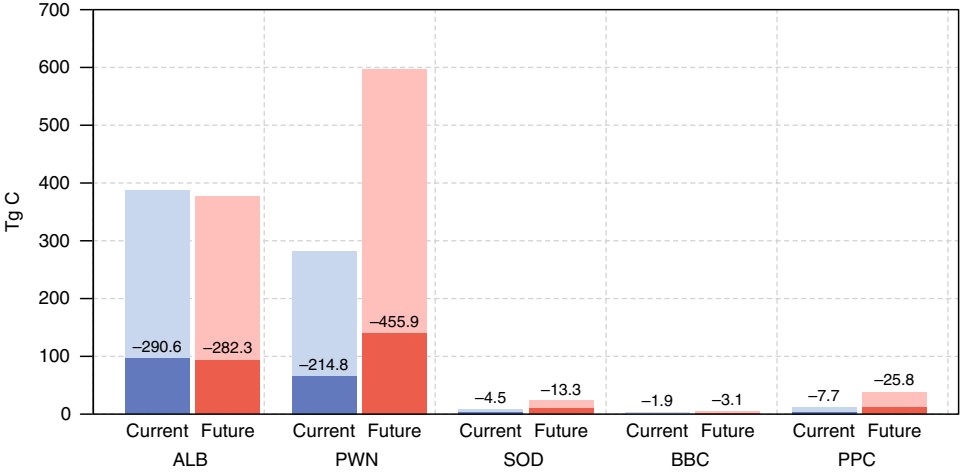

**Fig. 3** Effect of pest management on live tree carbon at risk under current and future climate. The total height of the bars indicates the C at risk without pest management, with the light-colored portion of the bars illustrating the management effect, and the dark colored portion showing the residual live tree C at risk with pest management in place. Numbers indicate the values (in Tg C) by which C at risk can be reduced through pest management. Current refers to the climate conditions of 1950–2000, while future indicates values for the mid-twenty-first century (2030–2080) under an intermediate climate scenario (RCP 4.5). ALB: Asian Long-horned Beetle (*Anoplophora glabripennis*), PWN: Pine Wood Nematode (*Bursaphelenchus xylophilus*), SOD: Sudden Oak Death (*Phytophthora ramorum*), BBC: Beech Bleeding Canker (*Phytophthora kernoviae*), PPC: Pitch Pine Cancer (*Fusarium circinatum*)

required to better predict the impacts of and resilience to emerging novel forest disturbance regimes[12,43].

Alien pests may not only diminish regulating ecosystem services but can also impact provisioning, cultural, and supporting services of forests[44,45]. For PWN, for instance, the potential economic impact for Europe was estimated to 22 billion Euros[46]. Frequently, however, alien pests affect non-marketed ecosystem services even more strongly than marketed services[47]. Furthermore, the invasion of alien pest species threatens the biological diversity of forests[14]. It is thus of crucial importance that rigorous measures are taken to halt or reduce the spread of alien pests. As all five species studied here are currently found in limited areas of Europe only, halting their spread and aiming for their eradication is a key priority for management. For PWN, for instance, a quarantine area has already been established in the infested region of Portugal, surrounded by a buffer zone in which all host trees have been removed, and export of pine wood products from the infested region is prohibited[48]. Also proactive management measures such as thinning and prescribed burning can reduce the pressure from certain pests[49,50]. Long-term silvicultural measures such as increasing diversity at different scales (e.g., genetic diversity, tree species diversity, structural diversity, diversity at the landscape scale) could also help to increase the resistance and resilience of forests and their ecosystem services to alien pests[51–53]. We conclude that alien pests are a major threat to Europe's forests, endangering their current role as important C storage. Consequently, increased efforts are required in forest policy and management to halt the introduction and spread of alien forest pests.

## Methods

**Potential distribution of alien forest pest species**. We combined high-resolution maps of forest C stocks and net primary production (NPP) in Europe, derived from forest inventory data and remote sensing, with species distribution modeling and C impact modeling to assess the potential distribution of alien forest pest species under current climate and climate change, and to quantify their impacts on the forest C cycle. Supplementary Fig. 9 gives an overview of our analysis framework.

We used four criteria to select focal invasive alien pests for our analysis: First, the pest must be alien in Europe's forests, second, have widely distributed hosts among the native tree species of Europe, third, cause severe impacts on their host species (i.e., tree mortality), and fourth, must be already introduced in Europe in spatially restricted initial populations. From the species that best met these four criteria, we chose five agents to cover a broad variety of taxonomic groups as our

study organisms, i.e., ALB (*Anoplophora glabripennis*), PWN (*Bursaphelenchus xylophilus*), SOD (*Phytophthora ramorum*), BBC (*Phytophthora kernoviae*), and PPC (*Fusarium circinatum*). Species distribution models (SDMs) were used to quantify the climatically suitable areas of the five alien pest species in Europe under current climate (1950–2000) and scenarios of future climate (2030–2080).

Data on the global distribution of the five forest pests were compiled from a variety of sources, most importantly the Global Biodiversity Information Facility database (http://www.gbif.org/), the EPPO Global database (https://gd.eppo.int/), and the Invasive Species Compendium database (http://www.cabi.org/isc/). Multiple occurrences within 10' ×10' grid cells and clearly erroneous records, i.e., those in water bodies, were removed. Owing to an apparent lack of data on species occurrences in Asia, we also considered coarse-scale data sources from the area to fill this gap. Furthermore, we also included occurrences from the alien range of the pests because species can sometimes expand their realized climatic niches in their alien range[54–56]. Species occurrences and references for the five focal species are given in Supplementary Figs. 3–7.

For characterizing temperature and precipitation of current climate (period 1950–2000), we used four bioclimatic variables provided by WorldClim[57]: mean diurnal temperature range, minimum temperature of the coldest month, mean temperature of the warmest quarter of the year, and annual precipitation. These variables capture limits to species distributions set by (low) temperature extremes, available energy and water, and also include an indicator of seasonality. All climatic variables were available at a spatial resolution of 10 arc minutes. Possible future climates in Europe were represented by three emission scenarios of the IPCC CMIP5 scenario family: the moderate RCP 2.6, the intermediate RCP 4.5, and the severe RCP 8.5 scenarios (Supplementary Table 1)[58]. The respective downscaled monthly temperature and precipitation time series were retrieved from the CORDEX portal (http://cordexesg.dmi.dk/esgf-web-fe/live, Supplementary Table 1). From these climate time series, we recalculated 10' resolution maps of the above described climate variables for all scenarios. A 50-year average over the period 2030–2080 was used to represent the climate of the mid-twenty-first century in the analysis.

We used the biomod2 platform[59] in R[60] to model the climatic niches of species and subsequently project their potential current and future geographical distribution. Niche models were parameterized by relating documented species occurrences to WorldClim climate data for the reference period 1950–2000. From the modeling algorithms available in biomod2, we selected Generalized Linear Models (GLM), General Additive Models (GAM), Boosted Regression Trees (BRT), and Random Forest (RF). As we only had information on species occurrences but no data on absences, we generated pseudo-absences following the recommendations of Barbet-Massin et al.[61]. Specifically, we used 10,000 randomly distributed absences for regression models (GLM and GAM). For machine-learning models (BRT and RF), we chose the number of pseudo-absences equal to the number of occurrences reported in the data, randomly situated in a radius of 200 km around the occurrences[61]. In the latter case, pseudo-absence generation, and hence model calibration, was repeated 10 times per species to ensure that the selected pseudo-absences did not bias the final predictions. For all models, the weighted sum of pseudo-absences was set to equal the weighted sum of presences. The predictive performance of the models was evaluated by means of the true skill statistic (TSS)[62] based on a repeated split-sampling approach (three replicates) in which models were calibrated with 80% of the data and evaluated against the

**Table 3 Equilibrium C cycle effects of a potential invasive alien disturbance regime compared to the natural disturbance regime in Europe**

| | | Current climate (Tg C) | Future climate (Tg C) |
|---|---|---|---|
| Invasive alien disturbance regime | ALB—Asian Long-horned Beetle | 246.0 | 252.0 |
| | PWN—Pine Wood Nematode | 188.4 | 291.2 |
| | SOD—Sudden Oak Death | 9.0 | 32.7 |
| | BBC—Beech Bleeding Canker | 5.7 | 11.7 |
| | PPC—Pitch Pine Canker | 10.4 | 46.5 |
| | All | 308.7 | 392.6 |
| Natural disturbance regime | Wind, native bark beetles, and wildfires | 319.8 | 503.4 |

Values indicate the long-term reduction of total ecosystem C storage capacity in Europe's forests due to disturbance (Tg C). For invasive alien pests, the implementation of effective pest management measures is considered under both current climate (1950–2000) and future climate (RCP 4.5, 2030–2080), as also natural disturbance risk is commonly managed in Europe's forests. Values for the natural disturbance regime of Europe are taken from Seidl et al.[6] and refer to observations for 1971–2010 (current climate) and the median projection for an ensemble of 12 climate change scenarios for 2021–2030. Please note that, while methodologically similar, the reference periods and climate scenarios differ between the assessments of invasive alien and natural disturbance regimes. All: upper bound of the equilibrium C cycle effect from all five invasive alien pests jointly

remaining 20% (Supplementary Table 8). The final models were used to make two independent ensemble projections of the potential spatial distribution of the five forest pests under current climatic conditions and three climate change scenarios, where one ensemble was based on the two regression techniques (GLM and GAM) and the other on machine-learning approaches (BRT and RF). A binary output of each ensemble projection was generated based on a threshold selected to maximize the TSS score[63,64]. The final climatically suitable area of a pest species was determined by aggregating the two ensembles to a consensus prediction, i.e., areas were only classified as climatically suitable if both ensembles agreed on the potential presence of a species in a given cell.

**Live tree C stocks of Europe's forests**. Assessing the C at risk from alien pest species and identifying potential future hotspots of C impact requires a spatially explicit estimate of the C currently stored in Europe's forests, as well as a distinction between C in host tree species and C in non-host tree species. We estimated live tree C stocks from continental-scale terrestrial forest inventory data and satellite-based remote sensing. Specifically, the spatially explicit C data set employed here was developed using a large data base of national forest inventory plots (containing >200,000 plots) in conjunction with newly downscaled European climate data and auxiliary spatial data sets. These data were combined to derive wall-to-wall maps of mean live tree C per hectare of forested areas at 8 arc minute resolution, using cluster analysis and nearest neighbor imputation[19,65,66]. The thus derived C density includes all tree species assessed by the forest inventory systems in Europe and contains all trees larger than the applied inventory diameter threshold (usually 5 cm, measured at a tree height of 1.3 m). For more details on European-scale C mapping, we refer to previously published work[19,65,66]. To derive C content from C density, the forest cover per cell from a pan-European forest cover map[67] was used, and C estimates were resampled to match the 10 arc minute resolution of SDM data (Supplementary Fig. 10).

For our analysis, it was additionally important to determine the share of C that is contributed by potential host tree species of the five invasive alien pests (Table 1). This potential host C was derived for each cell in a two-stage approach, combining species information from forest inventory and remote sensing. First, we used forest inventory data to calculate live C shares for seven tree species groups determined by life history traits (i.e., light-demanding conifers, shade-tolerant conifers, Mediterranean conifers, fast growing deciduous, light-demanding slow-growing deciduous, shade-tolerant slow-growing deciduous, and evergreen broadleaves) for each grid cell. Second, we derived the C relevant for each pest species by further sub-setting these groups based on relative tree species shares of the European Species Map (ESM). The ESM provides relative shares of 20 tree species at 1 km resolution for Europe, derived from forest inventory data and remote sensing[68]. Supplementary Fig. 11 shows the resulting maps of live tree C in host tree species susceptible to alien pest species.

**C impact modeling**. We assessed the potential impact of forest pests in three sequential steps, quantifying, first, the potential live tree C at risk from a complete invasion of all regions climatically suitable for the pest, second, the C recovery time, i.e., a measure of C cycle consequences of such an invasion, indicating the time needed for the system to recover the potential live tree C lost via net primary productivity, and third, the equilibrium C cycle effects of novel pests, i.e., how much the undisturbed C storage potential of forest ecosystems in Europe is reduced by alien pests, taking into account disturbance severity, the fate of the disturbed C, the C residence time of the system, as well as the potential of the forest to recover (Supplementary Fig. 9). While steps one and two focus on live tree C as response variable, step three quantifies disturbance effects on total ecosystem C. The analyses in all three steps are described in detail below.

We calculated the potential live C at risk by combining the potential distribution of each pest species (Fig. 1) with the C stocks of its respective host tree species (Supplementary Fig. 11). C at risk was derived from the C in host tree

species in a cell climatically viable for the pest multiplied by a species-specific severity, i.e., the rate of host tree mortality that can be expected in an infested area (Table 1). The abundance of the pest and its potential effect on severity was not explicitly considered. As all pest species addressed here do attack small trees only rarely[21,22], we excluded trees with a diameter at breast height <5 cm from our analysis. C at risk was calculated individually for each pest species. To estimate the upper bound of live tree C at risk from all five species taken together, we calculated a cell-level maximum value by, first, separating coniferous and broadleaved hosts, second, calculating the maximum impact from among the five pest species for each of these two tree species groups, and third, aggregating them additively. This upper-bound estimate of live C at risk accounts for the fact that some pests considered here depend on the local availability of the same host tree species (e.g., PWN and PPC) and thus cannot be aggregated additively, while others can have additive effects by potentially infesting different tree species groups at the same location (such as ALB and PWN). Beyond competition for the same host, additional interaction effects between the five pest species were not considered.

To elucidate the uncertainty of the thus estimated live tree C at risk, we calculated bootstrapped uncertainty intervals. We considered two sources of uncertainty in this analysis, the within-grid cell variation in C stocks as estimated from forest inventory data[19,65], and the range of mortality rates reported in the literature for each pest species. We sampled 100 C stock values based on the inventory-based standard deviation of C stocks for each grid cell (Supplementary Fig. 12) and drew 10 pest-specific severity values from a uniform distribution covering the reported range in the literature (Table 1). Uncertainties were quantified as the range between the 2.5th and 97.5th percentiles from the resulting set of 1000 estimates of live tree C at risk. This uncertainty assessment was conducted under current and future climate (scenario RCP 4.5) without considering the effect of pest management.

In a second step, we estimated the time needed for the system to recover the live tree C potentially lost from an invasion of alien pests through primary productivity. Based on the theoretical work of Weng et al.[20], we calculated C recovery time as (Eq. 1)

$$R = -\tau_1 \cdot \ln\left(\frac{f \cdot X_1 - U \cdot \tau_1}{X_{1,0} - U \cdot \tau_1}\right) \quad (1)$$

with $R$ the recovery time in years, $f$ the fraction of the pre-disturbance live biomass pool $X_1$ (t ha$^{-1}$) that indicates a full recovery, $U$ the annual NPP (t ha$^{-1}$), $X_{1,0}$ (t ha$^{-1}$) the C in the live biomass pool immediately after a disturbance by a pest, and $\tau_1$ the residence time of C in the live biomass pool (years). We assumed full recovery of live tree C stocks when a previously infested forest again reached 99% of its current C level (i.e., $f = 0.99$ in Eq. 1), and report $R$ for this level of recovery. NPP information was derived at 1 km horizontal resolution, based on remotely sensed vegetation information (MOD17) and high-resolution daily climate data[66] (Supplementary Fig. 13). The MOD17 algorithm employs key biogeochemical principles such as a light-use efficiency model and temperature-dependent autotrophic respiration estimates to provide consistent wall-to-wall NPP estimates at the continental scale. Previous analyses showed that the NPP data set used in this study (MODIS_EURO) agrees well with reference data from European forest inventory plots across scales and environmental gradients[66]. Residence times were estimated from country-scale forest inventory data[6] and spatially interpolated for each grid cell (see Supplementary Fig. 14). Recovery times were subsequently analyzed at the level of biomes for Europe (see Supplementary Fig. 15).

The potential effects of pest management on live tree C at risk and C recovery time were estimated using the same approach as outlined above but considering reduced mortality rates, as derived from successful examples of the management of our focal pest species (Supplementary Table 5). Our assessment of pest management effects thus does not account for a potentially decreased distribution of pest species (e.g., as a result of successful containment measures) and only includes the effects of reduced pest-induced tree mortality. We quantified the

potential effect of pest management based on experiences gained in managing the studied species in Europe and on other continents affected by the pests. We note that the effectiveness of pest management is contingent on available management measures and their implementation, the efforts and resources available to pest management, as well as the local context. Mortality rates may thus vary in space and time around the average values assumed here.

In the final analysis step, we compared the equilibrium C cycle effect of alien pests to that of the three agents currently dominating the disturbance regime in Europe's forests, i.e., wind, native bark beetles, and wildfire. Seidl et al.[6] estimated the equilibrium C cycle impacts of these native agents based on the theoretical C and disturbance model REGIME[20]. We replicated their approach in order to consistently compare the C cycle effects of the natural and potential novel disturbance regimes in Europe. In REGIME, the disturbance impact on the equilibrium C storage capacity is calculated as (Eq. 2)

$$X = U \cdot \tau_E \cdot \frac{1}{1 + \sigma \cdot \tau_1} \qquad (2)$$

with $X$ the expected total ecosystem C storage, $U$ the net primary productivity (Supplementary Fig. 13), $\tau_1$ the residence time of C in the live biomass pool affected by disturbance, $\tau_E$ the residence time of total ecosystem C (where $\tau_E = \tau_1 + \tau_S$, with $\tau_S$ the residence time of C in dead organic matter), and $\sigma$ the disturbance rate. In Eq. 2, the expression $U \cdot \tau_E$ represents the equilibrium C storage in the absence of disturbance, and $\frac{1}{1+\sigma \cdot \tau_1}$ represents the impact of disturbances (here: invasive alien pest species). $\tau_S$ was estimated for a combined detritus and soil organic matter pool $X_S$ (Eq. 3):

$$\tau_S = \frac{X_S}{\eta_1 \cdot U} \qquad (3)$$

with $\eta_1$ denoting the fraction of NPP entering the detritus and soil pools. Data for both the state variable $X_S$ and the biomass removal fraction (i.e., $1 - \eta_1$) were taken from a previous continental-scale assessment of disturbance effects on forest C storage[6]. Disturbance rate was derived as (Eq. 4):

$$\sigma = \frac{D}{MRI} \qquad (4)$$

with $D$ the amount of C potentially affected by a pest relative to the total live C stock, and MRI the mean disturbance return interval. We assumed that, once present, the frequency of an attack by a pest species is limited by the time needed from infection to tree death ($K$, see Supplementary Table 5) as well as by the ability of the forest to recover the lost C ($R$, Eq. 1). We consequently set MRI=$K$+$R$. The thus derived equilibrium C cycle effect of invasive alien pests was compared to the values estimated by Seidl et al.[6] for the current and future natural disturbance regime in Europe's forests.

It is important to note that live tree C at risk, i.e., the response variable in steps one and two described above, is a measure of the potential ecological impact of invasive alien pests, and does not directly equal C lost from the ecosystem. The equilibrium C cycle effect calculated in step three, on the other hand, represents a more comprehensive estimate of the potential long-term reduction of forest C storage potential resulting from a complete invasion of alien pests. It accounts for continental-scale variation in C residence time (reflecting differences in forest age structure and management history[69]), considers rates of extraction of killed biomass (see Supplementary Table 9 for a sensitivity analysis), accounts for a transfer of dead organic material to detritus C pools (Eq. 3), and includes spatial differences in recovery of C through primary production (see also Supplementary Fig. 9). All analyses were done at 10 arc minute spatial resolution for continental Europe (here delineated by the 28 countries of the European Union as well as Norway, Switzerland, Liechtenstein, Bosnia and Herzegovina, Macedonia, Serbia, Montenegro, and Albania), using the R Project for Statistical Computing[60].

**Sensitivity analysis of ensembles of future species distribution**. As statistical distribution modeling of invasive species is inherently uncertain, we used a consensus prediction based on two ensembles of different distribution modeling approaches. To further assess uncertainties we—in addition to the consensus prediction—analyzed the results of a more liberal model aggregation, in which occurrence predictions by only one of the two model ensembles were sufficient to indicate climatic suitability (i.e., using an OR operator to combine the two model ensembles). We, in other words, contrasted a consensus estimate of species distribution with a more liberal one (with the latter more closely conforming to a precautionary principle approach in risk assessment).

The results of this analysis showed that the climatically suitable area under liberal model aggregation was considerably larger than the consensus estimate (Supplementary Fig. 8, Supplementary Table 2). The highest uncertainty in predicting species distributions (i.e., the strongest disagreement between the two ensemble models) was found for PWN, where the climatically suitable area was +235.2% larger under liberal model aggregation compared to the consensus estimate (current climate). Disagreement between ensemble predictors was lowest for ALB, for which the liberal model aggregation increased the climatically suitable area by only +17.4% compared to the consensus prediction. Please note that all results reported in the main text pertain to the consensus prediction.

**Sensitivity analysis of changing forest composition and functioning**. We based our assessment of the potential impact of invasive alien pests on high-resolution information of current forest composition and functioning. To assess the effect of potential future changes in C uptake and storage as well as in tree species composition on our findings we conducted sensitivity analyses. Based on projections of future forest trajectories, an increase in C stored in live biomass can be expected for Europe's forests. Depending on the future management policies, this increase has been estimated to range from 10.9% to 19.9% by 2030[70]. To assess the effect of potential increases in Europe's C stock on C at risk from invasive alien pests, we thus reran our analyses with C levels elevated by +10% and +25% in a sensitivity analysis.

While the expected future changes in C stocks are mostly an effect of the current age class structure of Europe's forests[4,69], also C uptake is expected to change in the future, in response to a variety of global change factors. However, NPP changes are likely to be highly variable in space[71]. Furthermore, considerable uncertainties exist regarding the persistence of a $CO_2$ fertilization effect[72]. Reyer et al.[72], for instance, estimated mid-century NPP changes to range from $-1.24$ to $+1.11$ Mg C ha$^{-1}$ year$^{-1}$ (no $CO_2$ fertilization effect), and from $+0.13$ to $+2.55$ Mg C ha$^{-1}$ year$^{-1}$ if a persistent $CO_2$ fertilization effect is assumed. As NPP is crucial for the recovery of ecosystem C after a disturbance, we analyzed the effect of changing NPP by varying current values by ±15% in our sensitivity analysis.

Finally, also tree species composition is likely to change in the future. However, while SDMs project rapid shifts of species envelopes[73], more realistic assessments of transient changes in tree species until the mid-twenty-first century predict only small changes in species composition[41,70]. UNECE and FAO[70], for instance, project tree species changes (relative to growing stock, which is closely correlated to live tree C) until 2030 to be <1%. Also Thom et al.[41] report changes of <5% for the first half of the twenty-first century in their landscape-scale analysis. To assess the effect of potentially changing host tree species abundances, we varied the proportion of C in host tree species on the total C by ±5 percentage points in our sensitivity analysis. The response variable for the sensitivity analysis was C recovery time, assessed jointly for all five pest species. The analysis was conducted as a local sensitivity analysis, i.e., varying one factor at a time.

Of the three factors analyzed, a strong increase in C would have the highest impact on the C cycle effect of forest pests (Supplementary Table 7). Conversely to changes in C stocks, the recovery time responded strongly nonlinearly to changes in NPP and the C share in host tree species. NPP decreases, for instance, increased the C recovery time disproportionally more than NPP increases. Furthermore, a relative increase in the host tree species share of pests by five percentage points elevates the C recovery time by +15.8%.

**Sensitivity analysis of changing C residence times and salvage assumptions**. To further elucidate the sensitivity of C cycle impacts with regard to key model parameters, we tested the effect of changes in C residence time and salvaged on the equilibrium C cycle effect calculated via Eq. 2. C residence times were found to vary considerably across Europe (cf. Supplementary Fig. 14). This variation represents, among other things, the considerable variation in stand ages and standing timber volumes in Europe at the continental scale[69]. Increasing the live C residence time also increased the simulated impact of alien pests on the C cycle. The response of the equilibrium C cycle effect of disturbances to changes in live C residence time were close to being proportional, with a 10% parameter variation resulting in a 9% response (Supplementary Table 9).

Another important consideration for the C effect of disturbances is how disturbed areas are treated by management. In Europe, disturbed areas are frequently salvage harvested[74,75], with the removal of trees killed by invasive pests often being mandatory due to phytosanitary reasons. We thus assumed a complete removal of the trees affected by alien pests in our calculations of their equilibrium C cycle effect. However, in order to test the effect of different levels of salvage harvesting we reduced the salvage fraction from 1.00 to 0.80 and 0.66 in a sensitivity analysis. As expected, the results show that lower levels of salvage harvesting reduced the equilibrium C cycle impact of invasive alien pests. However, the effect was not proportional, with a reduction of salvage harvesting by 34% reducing the C cycle impact by 22.6% under current climate (Supplementary Table 9).

**Data availability**. Data on potential pest species distribution, live tree C in host tree species, C at risk, and C recovery times are available in the online repository Figshare (https://doi.org/10.6084/m9.figshare.5999243). All other data sets used are available from the respective references.

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

## Acknowledgements

We acknowledge funding from the Austrian Science Fund though START grant Y895-B25 (to R.S., W.R.) and through the project "Who Is Next", grant I-1443-B25 (to G.K., F.E., S.D.).

## Author contributions

R.S., F.E., and S.D. developed the idea and designed the research. F.E. and G.K. compiled the data on forest pests. G.K. and S.D. developed and ran the species distribution models. A.M. and M.N. contributed data on continental-scale forest carbon uptake and stocks. W.R. and R.S. performed the carbon impacts modeling and analyzed the data. R.S. wrote the first draft of the text. All authors contributed to writing and revising the text.

## Additional information

**Competing interests:** The authors declare no competing interests.

