## [Peer Review File · Nature Communications]

Reviewers' comments:

Reviewer #1 (Remarks to the Author):

This paper describes the likely consequences of current and future invasions by a suite of different non-native forest pest species across Europe on changes in live forest carbon (C). The authors use a combination of advanced spatial distribution modelling approaches, forest inventory plot data, remote sensing information, and forest C modelling under current climate conditions and three future climate change scenarios to predict the future impacts of these pests alone and in total on live forest C, and C recovery to pre-invaded levels. In addition, the authors also test the effectiveness of current management methods for each of the five pest species to mitigate their impacts of live forest C. Overall the authors present clear and compelling case that invasion by forest pests will drive pervasive declines in live forest C; that few species can drive this decline; that management can, at best, only partially mitigate these declines; and finally, that the future impacts of these pests will worsen with climate change because of greater suitable climate available for these invaders. This paper provides an important benchmark for forecasting the likely effects of major forest pest species across Europe, is both data rich and compelling, and has global implications. Overall the paper is well-written, clearly presented and topical; I hope my comments help to strengthen it:

-Overall the manuscript is well written and clear. However, the actual text of the main text is 'flat' and in some cases unclear (e.g.,). I recommend a careful technical edit of the text.

-Some further justification for your pest species selection (i.e., expand lines 49-51). Are there many other candidate pest species, or widely-distributed but not economic tree species that are likely to be affected?

-lines 63-64. A subset of natural disturbances were considered here. Are these, or other disturbances, likely to shift with predicted changes in climate? (i.e., these disturbances act as your baseline against which pest impacts are evaluated, but what is the evidence for this baseline shifting across Europe in the coming decades?).

-lines 86-90. Would co-invasion by 2 or more pest species dampen or amplify these impacts? Likely, but my understanding is the way the models were constructed do not allow evaluation of impacts of co-occurring pests (see also lines 379-385). I'm happy to be shown wrong on this; but respectfully request additional clarification in the text.

-A brief mention of the magnitude/importance of pest-driven declines in live forest C could be made in the paper; for example, what is the change in forest live C expected over this period as a result of other climate change (drought, fire) or recent land use change?

-Does reinvasion negate the effectiveness of management? In other words, is pest management only effective as the invasion proceeds, but is negligible once the pest species have already invaded.

-It seems like the model assumes maximum impacts of pest species on arrival, but what is the evidence for density or abundance (vs. presence) effects of these pest species? This is a generalisable phenomenon with invasions (e.g., Parker et al. 1999 and more recent derivatives of invader impacts in which abundance is an explicit term).

-Similarly (lines 373-374 and elsewhere) occupancy is driven by climatic viability of cells predicted by four worldclim variables here. Is viability related to likely abundance, and hence potential impact, differently across the contrasting pest species being considered here. I assume this would, in practice, just add another source of variability for tree mortality (e.g., lines 414-415). Some clarifying text is all that is needed here.

-lines 356-360. What was the purpose for creating tree species groups here, and then subdividing these further into species (lines 361-)?

-the "cons" and "lib" models look reassuringly similar overall with one exception PWN in suppl Fig. 11; is there an easy explanation for this?

-lines 132-133 of suppl: ref here for trajectories?

-I thought it might be worth emphasising more strongly that the value (sensu lato) of management increases substantially even with modest climate changes (Suppl. Fig 14).

Reviewer #2 (Remarks to the Author):

Invasive alien pests threaten the carbon stored in Europe's forests

Rupert Seidl, Günther Klöner, Werner Rammer, Franz Essl, Adam Moreno, Mathias Neumann, and Stefan Dullinger.

The authors present an analysis of the potential impact of five invasive forest pests and pathogens on live carbon biomass in European Forests, using climate models and information on species ranges and tolerances to project how climate change could affect live tree biomass and subsequent recovery times in forested areas.

Overall, the analyses are sound and the authors use appropriate climate scenarios, recent forest inventory data for baseline tree biomass, and MODIS derived estimates of net primary productivity to derive recovery times. The manuscript is well written and appropriate for Nature Communications.

Major concerns

One concern with the analyses are that the pest and pathogen range information does not seem to have an upper temperature bound. Perhaps there is little information on this for the five selected species, but if for example, Asian longhorn beetle behaves like Southern Pine Beetle in the SE US, warmer temperatures seem to correlate with lower activity of SPB in the now warmer part of the range of SPB. The authors also do not consider that pathogens and parasitoids of some of these species could also increase with changing climate. The authors wouldn't necessarily change any of their analyses, but should mention these factors as potential sources of uncertainty in the Discussion section.

A second major concern is that throughout the manuscript, the authors focus on live tree biomass, but this misses an important point about insect and pathogen damage, because tree mortality produced snags and coarse woody debris. Decomposition rates estimated with a k value on the order of 0.05 to 0.10 lead to long term (relative to live tree biomass) C storage, eventually on the forest floor and in soil.

As many of these forests are intermediate in age and few are likely to be old growth, then coarse woody debris is a relatively new feature in these forests. We have found that CWD derived from

invasive insect damage is enhancing ecosystem respiration, and although gross ecosystem productivity and NPP (as the authors use) can recover rapidly following infestations, the overall net flux of C is lower in insect damaged forests because of the higher respiration rates.

The authors should at least mention this, because their main point about forest C storage is missing some relevant processes. They also contrast insect and pathogen damage with fire and should probably include forest harvesting, and should again note that these disturbances have very different effects on the fate of detrital C.

Specific concerns

Lines 20-21: "...alien pests on the forest C cycle remains unresolved". We do know a lot about invasive species and forest C; it is the interaction with climate and invasive species that is uncertain.

Lines 32-33: Is it relevant to this set of analyses that C forests exceed the amount in the atmosphere? It seems that it is the rate of change of both pools is of greater importance.

Line 36; "...has removed many dispersal barriers for species"

Lines 47-55: The selection of studied species is interesting, because of the comparison/contrast between very different taxa. Do the authors consider these to be the most active/aggressive pests and pathogens? Why these species were selected could be justified further by a rewording of a sentence or two.

Lines 86-93. When the authors consider risk to forest C in this section of the Results, they are not considering the fate of the detrital C (or of salvage logging). This forest C doesn't disappear, so the way this section is written seems to overinflate the impact of the 5 species, as per the major comment above.

Line 97. The wording "theoretical model of C dynamics" is odd. The underlying algorithms for the MODIS NPP calculations are derived from Biom BGC, so it might be better to say "process-based model" or "mechanistic model"

The approach the authors use to calculate recovery times is effective. However, in this analysis, do the authors consider stand age? This has been a more difficult factor to tease out from the MODIS and remote sensing data, because predicted NPP is mostly driven by estimated leaf area from NDVI and some climate data. This seems relevant to the average time to recovery that is presented in Figure 2, because for example Mediterranean stand are younger on average than temperate stands.

Also, Figure 2 is essentially a “worst case scenario” and it would be helpful to remind the reader that this is not only “all areas suitable” for invasion but also represents some range in the percentage of tree mortality in the legend.

Lines 136-137. This almost seems like too strong of a statement, and also ignores many of the other services that European forests have provided. Perhaps reword to encompass a broader scope?

Lines 146-154. This is where the potential impact of higher temperatures and pathogens and predators of the five invasive species could be mentioned. It is set of uncertainties that could affect their impact to forest carbon storage and productivity.

Lines 155-156 and 176-177. This paragraph would benefit from a bit more detail about the fate of C during disturbance. If the authors actually consider “forest C storage” as they state, then they have to include C in dead trees and detritus derived from these contrasting disturbances. If they did this, the times to recovery presented in Figure 2 would be much shorter, because then NPP needs only to catch up to the rate of CWD decomposition.

Lines 170-173 and Table 1. The authors largely mention reactive management strategies to these pest and pathogens, but some proactive forest management treatments such as thinning and prescribed burning have shown to be effective at reducing the impact of some invasive species (although not the five that authors consider). The authors should consider mentioning this, especially for ALB and PWN.

Table 1. The ranges of mortality for ALB and PWN seem quite high. Does this include any management activities? “Quarantining infested areas” is true, but if the authors could provide more specific treatments, that would help inform forest managers. The addition of “not selling infected wood or moving firewood” or other more specific management practices would be helpful where appropriate.

Most of the methods are sound, and the authors should be commended for the integration of many sources of information in this synthesis. It would be interesting to follow up on some of the species distribution models (both pests and trees) to detail where any high temperature effects occur, as mentioned above.

Carbon stocks of Europe's forests: The methods the authors use to estimate C in live tree biomass is sound, and field data from the forest census plots are essentially equivalent to USFS Forest Inventory and Analysis inventory data collected in the United States. Do these inventory data also include dead trees and CWD on the forest floor? This is typically sampled less frequently, but if the authors were to estimate true forest C storage, these data should be included, as per comments above.

Lines 353-365. How accurate were the attributed tree species groups to the actual species composition in the field inventory data? This is likely beyond the scope of this manuscript, but it does have some bearing on the estimate of mortality, especially for pines vs. other conifers.

The methods used to estimate live biomass accumulation (NPP) using MODIS data seems sound. The simple equation used to calculate C recovery time is adequate. Where do the estimated residence times of C in live biomass come from? This is where it would be useful to know two parameters; the spatially averaged age of forests in the simulation unit, and the maximum age and biomass of each tree species in each simulation unit. Because many of these forests are intermediate in age, they likely have not come close to reaching maximum age or biomass.

Lines 404-410. As the authors state, they do not account for management practices in reducing the distribution of pest species. However, these have shown to be effective in some cases. It seems like this would be a fruitful area for sensitivity analyses for forest mortality, especially for ALB and PWN.

Lines 433-439. This is a key point and should be elevated to the Discussion section.

Reviewer #3 (Remarks to the Author):

Seidl et al. use species distribution modeling (SDM) techniques to tackle an ambitious, very important problem. That is, providing estimates of potential changes to forest carbon in response to the invasion of pests and pathogens following climate change. Here I have commented on their SDM methods, as requested.

The methods are generally well written and likely accessible to a broad audience across multiple disciplines.

Application of species distribution models to study effects of global environmental change has exploded in popularity over the past decade (See multiple review articles by W. Thuiller, Janet Franklin, and others). A common challenge discussed in this literature is the tradeoff of using simple correlative approaches (as done in this manuscript) for parameterizing static SDMs versus more complex mechanistic approaches that attempt to model dynamic processes of pest population growth and dispersal over larger regions (e.g. Cunniffe et al. 2016 PNAS).

The authors have appropriately carried out the simpler correlative approach, which require less data (and thought) on the species' life history strategies. But, they might consider justifying the strength and weaknesses of their approach versus using a dynamic modeling approach. This exercise should also involve stating a number of model assumptions about equilibrium, dispersal etc..., which aren't currently included in the manuscript but are recommended in Thuiller's and Franklin's best practices of SDM applications in environmental change studies. Siedl et al. only consider first order interactions between pest redistribution and climate change. For example, the possibility of host redistribution and changes to biotic interactions under novel climates are not considered. Other simplifying assumptions also influence the results. For example, do pests and pathogens really respond to annual precipitation amounts and 50-year climate averages? Probably not, a large literature exists showing how sensitive pests and pathogens are to the timing of specific weather conditions and seasonal variability in host phenology/susceptibility. Finally, dispersal rates for future invasion seem to be ignored. The authors should better explain their assumption of complete spread in the future time step.

Overall, the methods chosen are sound. There's no single way to implement SDMs in climate change studies. But, the assumptions and their limitations could be more thoroughly described.

Reviewer #4 (Remarks to the Author):

Comments on Seidl et al. Invasive alien pests threaten the carbon stored in Europe's forests

The paper evaluates the impacts of pest invasion on the carbon storage of European forests. The authors used a niche model to predict the potential distributions of the five major invasive pests for European forests based on current distribution data. And then, they estimated their effects on forest biomass carbon storage using an equilibrium disturbance-carbon storage model. This paper provides very useful baseline evaluation for the risks of forest carbon storage due to pest invasion resulted from climate change. The results are critical for forest management. The paper is well written and the methods are clear and solid.

I only have two minor suggestions:

1. For the Equation 1 that is used to calculate the recovery time, my suggestion is to refine the explanation of X_1 and f . I put the equation below for convenience:

$$R = -\tau_1 \ln \left(\frac{fX_1 - U\tau_1}{X_{1,0} - U\tau_1} \right)$$

This equation describes the recovery time after a pest attack event (i.e., a disturbance event). Since the authors assume the disturbance event will not change forest productivity (U) and background mortality (reciprocal of τ_1), it's OK to let $U\tau_1$ be the target of recovery. But, according to the model, X_1 is actually a sample of the random variable of forest biomass before the invasion of the alien pests. It's expectation (or mean) is:

$$E[X_1] = U\tau_1 \frac{1}{1 - \sigma_0\tau_1},$$

where σ_0 is the disturbance regime before the invasion of the alien pests. That's why the authors can get a reasonable recovery time even with $f=0.99$. So, if the authors let $f=1$ (then remove it from this equation), they still can get the similar recovery time while keeping this equation clear. Thus, it becomes recovering to the state before pest invasion.

Lines 102~103 in Page 4: "Recovery is slowest in boreal forests ... lower annual C uptake than ...". Here, I think it is because of the high residence time in boreal forests. For the system $\frac{dX}{dt} = u - \rho X$, the recovery time is a function of turnover rate ρ (i.e., the reverse of τ) because the fraction of recovery is:

$$\frac{X(t)}{u/\rho} = 1 - e^{-\rho t}.$$

2. If the invasive pests stay in these forests forever, rather than a single attack, the disturbance regime shifts from σ_0 to σ and the equilibrium forest biomass will be:

$$E[X_1(\sigma)] = U\tau_1 \frac{1}{1 - \sigma\tau_1}$$

This situation will permanently reduce the carbon storage of European forests.

But as for the disturbance regime after pest invasion ($\sigma = \frac{D}{K+R}$, Equation 3 in Page 19), I cannot figure out how it is derived. According to the REGIME model, $\sigma = \frac{D}{MRI}$, where D is the mean

ratio of the biomass removed by pest attack events and MRI is the mean return time of the events. How MRI equal to “K+R”? Please explain it.

Reviewer #1

This paper describes the likely consequences of current and future invasions by a suite of different non-native forest pest species across Europe on changes in live forest carbon (C). The authors use a combination of advanced spatial distribution modelling approaches, forest inventory plot data, remote sensing information, and forest C modelling under current climate conditions and three future climate change scenarios to predict the future impacts of these pests alone and in total on live forest C, and C recovery to pre-invaded levels. In addition, the authors also test the effectiveness of current management methods for each of the five pest species to mitigate their impacts of live forest C. Overall the authors present clear and compelling case that invasion by forest pests will drive pervasive declines in live forest C; that few species can drive this decline; that management can, at best, only partially mitigate these declines; and finally, that the future impacts of these pests will worsen with climate change because of greater suitable climate available for these invaders. This paper provides an important benchmark for forecasting the likely effects of major forest pest species across Europe, is both data rich and compelling, and has global implications. Overall the paper is well-written, clearly presented and topical; I hope my comments help to strengthen it.

Response: We thank Reviewer #1 for her/ his positive assessment and for the constructive suggestions on how to further improve our work. We have incorporated these into the revised version of the manuscript (see details below) and they have helped to further refine our manuscript!

Overall the manuscript is well written and clear. However, the actual text of the main text is 'flat' and in some cases unclear (e.g.,). I recommend a careful technical edit of the text.

Response: We have revised the text in this regard, and have aimed to increase the clarity in the new version of the manuscript. Also, we feel that by extending the text to meet the more

liberal word count and reference limits of Nature Communications we were able to clarify some of the issues that might have remained unclear in our initial submission.

Some further justification for your pest species selection (i.e., expand lines 49-51). Are there many other candidate pest species, or widely-distributed but not economic tree species that are likely to be affected?

Response: We agree with Reviewer #1 that the selection of our focal pest species warrants a better description – in fact this was also requested by Reviewer #2 (see below). We have now considerably extended the respective paragraph in the methods section, and explicitly name the four criteria that we used to select the five focal species (i.e., nonnative to Europe’s forests, widely distributed host tree species native to Europe’s forests, potential to cause severe impacts (i.e., widespread tree mortality), and being already introduced in a spatially restricted initial population in Europe). Please see the revised text in lines 252-261.

lines 63-64. A subset of natural disturbances were considered here. Are these, or other disturbances, likely to shift with predicted changes in climate? (i.e., these disturbances act as your baseline against which pest impacts are evaluated, but what is the evidence for this baseline shifting across Europe in the coming decades?).

Response: We thank the Reviewer for this valuable suggestion – indeed also the natural disturbance regime of Europe’s forests is changing distinctly in response to climate change. It might thus be interesting for the reader to compare the climate sensitivities of the natural disturbance regime to that of invasive alien pests. To illustrate this effect we have added a new results table (Table 2) in which we relate the equilibrium C cycle effect of invasive alien pests under current and future climate to that of current and future natural disturbances. With regard to the natural disturbance regime we here focus on wind, insects, and wildfire, which

are the three quantitatively most important forest disturbance agents in Europe (Schelhaas et al. 2003, Glob. Change Biol. 9, 1620-1633).

lines 86-90. Would co-invasion by 2 or more pest species dampen or amplify these impacts? Likely, but my understanding is the way the models were constructed do not allow evaluation of impacts of co-occurring pests (see also lines 379-385). I'm happy to be shown wrong on this; but respectfully request additional clarification in the text.

Response: It is correct that we did not consider potential amplifying interactions. A respective statement was added to the methods section (lines 177 – 181 and 378 – 380), to clarify this for the reader. With regard to dampening interactions we do account for the fact that a common host for multiple pest species can only be killed once. This is done by not aggregating the effect of the five pest species through naïve summation, but rather only considering the most severe agent for coniferous and broadleaved hosts respectively at the pixel level. This approach is described in detail in lines 370-380.

A brief mention of the magnitude/importance of pest-driven declines in live forest C could be made in the paper; for example, what is the change in forest live C expected over this period as a result of other climate change (drought, fire) or recent land use change?

Response: We agree that referencing the effects reported here to other drivers of forest change is high relevant, as it provides an important point of reference for the reader, helping to put our findings into context. A big problem in such comparisons often is that the large differences in underlying methodologies of study frequently turns this into an “apples vs. oranges” kind of comparison. However, as we here used the same methodology to quantify equilibrium C cycle effects as Seidl et al. (2014, Nature Climate Change 4, 806-810), we were able to directly compare our results to those of this previous effort focusing on disturbance agents native to Europe’s forests. We have now added a new table to the manuscript (Table 2)

in which the C cycle effect of invasive alien pests is directly compared to the effect of the native disturbance agents wind, bark beetles, and wildfire. This provides important additional information for the reader for contextualizing the C cycle effects reported here.

Does reinvasion negate the effectiveness of management? In other words, is pest management only effective as the invasion proceeds, but is negligible once the pest species have already invaded.

Response: The effect of pest management is here approximated by a reduced mortality rate of the agent. Consequently, the effect of pest management remains the same throughout the assessment, and does not change with time since invasion. This is described in detail in lines 400-406.

It seems like the model assumes maximum impacts of pest species on arrival, but what is the evidence for density or abundance (vs. presence) effects of these pest species? This is a generalisable phenomenon with invasions (e.g., Parker et al. 1999 and more recent derivatives of invader impacts in which abundance is an explicit term).

Response: It is correct that we do not simulate the continental scale population dynamics of invasive alien pests explicitly here. Hence we also are not able to estimate pest abundance, and consider potential abundance-dependent impacts in our analysis. This caveat was now added to the discussion (lines 177 – 181), and the suggested reference to Parker et al. (1999) was included, providing further context for the reader.

Similarly (lines 373-374 and elsewhere) occupancy is driven by climatic viability of cells predicted by four worldclim variables here. Is viability related to likely abundance, and hence potential impact, differently across the contrasting pest species being considered here. I

assume this would, in practice, just add another source of variability for tree mortality (e.g., lines 414-415). Some clarifying text is all that is needed here.

Response: We agree that this required some more clarification, and have revised the text accordingly (lines 410-411). In addition, we now also address this point in the discussion section, and have added a reference in order to provide more context for the reader (lines 177-184).

lines 356-360. What was the purpose for creating tree species groups here, and then subdividing these further into species (lines 361-)?

Response: Unfortunately, wall-to-wall information on tree species C for Europe is currently not available at the level of individual species. We thus operated with the tree species groups for which the data were available, and further refined them for our purpose as described in the text. Based on a comment of Reviewer #2 we have now also conducted further tests to evaluate our tree species C estimates, finding good correspondence of our tree species C maps and the reference values from terrestrial forest inventories at country scale (see our responses and Fig. R2 below).

the "cons" and "lib" models look reassuringly similar overall with one exception PWN in suppl Fig. 11; is there an easy explanation for this?

Response: There is not actually an 'easy' explanation. Under "cons" both the GLM/GAM and the GBM/RF models have to agree, whereas under "lib" the 'vote' of one of the two is sufficient to declare a site suitable for colonization. Differences between 'cons' and 'lib' hence indicate differences in predictions between the two model ensembles. In the case of PWN, there is disagreement between model ensembles with regard to the species' niche along several of the predictor variables, resulting in considerable geographical 'complementarity' among projections from individual models. As a result, the 'lib' projections extend over a

much broader area than the cons projections. The discrepancies concern tolerated temperature minima (coldest month mean) as well as energy requirements (warmest quarter mean) and drought resistance (annual precipitation). Overall, they suggest lower reliability of individual models in case of PWN (in comparison to the other species) as also indicated by lowest TSS values among all species (although they are still relatively high in absolute terms, Table S2). As this species is also one of the most influential ones with regard to carbon at risk, basing inferences on the consensus approach was deemed important for the robustness of our overall results. This decision may, however, result in a potential underestimation of its range, which is now also addressed in the revised and extended discussion section (lines 195-197).

lines 132-133 of suppl: ref here for trajectories?

Response: We have added two references reporting on the future C trajectories of Europe's forests here as suggested.

I thought it might be worth emphasising more strongly that the value (sensu lato) of management increases substantially even with modest climate changes (Suppl. Fig 14).

Response: We agree with Reviewer #1 that the potential effect of management to counter a climate-mediated spread of alien pests is an important finding of our study. We have now incorporated an additional Figure (Fig. 3) - explicitly displaying the management effect - into the main text of the manuscript, and correspondingly have extended the discussion of these results in the text.

Reviewer #2

The authors present an analysis of the potential impact of five invasive forest pests and pathogens on live carbon biomass in European Forests, using climate models and information on species ranges and tolerances to project how climate change could affect live tree biomass and subsequent recovery times in forested areas. Overall, the analyses are sound and the authors use appropriate climate scenarios, recent forest inventory data for baseline tree biomass, and MODIS derived estimates of net primary productivity to derive recovery times. The manuscript is well written and appropriate for Nature Communications.

Response: We thank Reviewer #2 for this overall positive assessment of our work.

One concern with the analyses are that the pest and pathogen range information does not seem to have an upper temperature bound. Perhaps there is little information on this for the five selected species, but if for example, Asian longhorn beetle behaves like Southern Pine Beetle in the SE US, warmer temperatures seem to correlate with lower activity of SPB in the now warmer part of the range of SPB.

Response: We thank Reviewer #2 for this comment and agree that considering upper thermal limits is important when making projections under climate change. Consequently, our SDM projections do in fact consider an upper temperature bound. All predictions are based on a consensus of two ensemble models, one comprising the combination of Generalized Linear Models (GLMs) and Generalized Additive Models (GAMs), and the other one the combination of Random Forests (RF) and Boosted Regression Trees (BRT). For all five species, the ensemble combining GLMs and GAMs fitted a unimodal relationship between probability of presence and temperature variables indicating both cold- and warm-limiting conditions (bio 6 – coldest month mean, bio 10 – warmest quarter mean). Probability of species presence hence decreases (sharply in most cases) under superoptimal temperature

conditions in the predictions. The ensemble combining RF and BRT did so for most, although not for all possible combinations of species and temperature variables. However, as occurrence of a species was only predicted where both models agreed on its presence (= the conservative consensus approach described in the main text and supplementary material), too warm temperatures necessarily resulted in absence predictions for all five species. Since the Reviewer explicitly mentions Asian longhorn beetle, we here exemplarily show the above described temperature response curves from our models for this pest species to illustrate the above described point (Fig. R1).

Figure R1: Modeled temperature response curves for *Anoplophora glabripennis* in the two model ensembles applied for prediction. Top panel: bio 6 - mean temperature of coldest month. Bottom panel: bio 10 - mean temperature of warmest quarter of the year. Please note that in the predictions a conservative consensus estimate of the two ensembles was used.

The authors also do not consider that pathogens and parasitoids of some of these species could also increase with changing climate. The authors wouldn't necessarily change any of their analyses, but should mention these factors as potential sources of uncertainty in the Discussion section.

Response: We have added the issue of pathogens and parasitoids of invasive alien pests to the discussion as suggested (lines 181-184 in the revised version of the manuscript).

A second major concern is that throughout the manuscript, the authors focus on live tree biomass, but this misses an important point about insect and pathogen damage, because tree mortality produced snags and coarse woody debris. Decomposition rates estimated with a k value on the order of 0.05 to 0.10 lead to long term (relative to live tree biomass) C storage, eventually on the forest floor and in soil.

Response: We agree that the fate of the effected C is important when assessing C cycle effects of pest species. We have thus computed two different values of C impacts, one being the live C at risk (which indeed does not account for decomposition and deadwood pools), and the other being the equilibrium C cycle effect. The latter does in fact account for the aspects mentioned by the Reviewer, considering continental-scale variation in C residence time (reflecting differences in forest age structure and management history), assuming certain rates of extraction of killed biomass, accounts for a transfer of dead organic material to detritus C pools, and includes spatial differences in recovery of C through primary production. These processes are considered using a C cycle model developed previously and described in detail in Weng et al. (2012, J. Geophys. Res. 117, G03014). We have now revised and extended the description of the methods to make this aspect more clear to the reader. Specifically, we contrast our two methods to estimate C impacts in lines 434-443 of the revised version of the manuscript. Nonetheless, owing to the large spatial scale that is addressed here, our full C cycle model is necessarily simpler than detailed approaches applied

at the level of individual stands. To further investigate the sensitivity of our approach we have thus added sensitivity analyses with regard to crucial parameters such as the extraction of deadwood to the revised Supplementary Material (see Supplementary Table S7).

As many of these forests are intermediate in age and few are likely to be old growth, then coarse woody debris is a relatively new feature in these forests. We have found that CWD derived from invasive insect damage is enhancing ecosystem respiration, and although gross ecosystem productivity and NPP (as the authors use) can recover rapidly following infestations, the overall net flux of C is lower in insect damaged forests because of the higher respiration rates. The authors should at least mention this, because their main point about forest C storage is missing some relevant processes. They also contrast insect and pathogen damage with fire and should probably include forest harvesting, and should again note that these disturbances have very different effects on the fate of detrital C.

Response: We agree with Reviewer #2 on the importance of the fate of the disturbed C, as well as on the relevance of changed ecosystem rates after disturbance. However, due to the continental scale of our analysis we were unfortunately not able to address all these aspects explicitly here (see also our previous response above). Nonetheless, issues like continental-scale differences in age structure are implicitly included in our model, via spatial differences in C residence times. In order to illustrate these better for the reader we have added a map of the residence times used to the Supplementary Material (Supplementary Figure S13). With regard to the issue of extraction, we'd like to point out that our modeling did in fact explicitly consider the fate of the live C affected by a disturbance agent, i.e. whether it remained on site or whether it was extracted through salvage logging. Generally, forests are managed with high intensity in Europe and salvage logging is common after disturbance (and even legally required in a number of European countries), which is why we selected a high salvage rate in our calculations. However, to further elucidate the effect of this decision we have now

conducted a sensitivity analysis of the effect of extraction on the equilibrium C cycle effect (see the newly added Supplementary Table S7). Furthermore, the description of methods was revised to more clearly describe our approach for the reader (lines 352-447).

Lines 20-21: "...alien pests on the forest C cycle remains unresolved". We do know a lot about invasive species and forest C; it is the interaction with climate and invasive species that is uncertain.

Response: We have revised the sentence in order to make clear that we here refer to future impacts of invasive pests under climate change.

Lines 32-33: Is it relevant to this set of analyses that C forests exceed the amount in the atmosphere? It seems that it is the rate of change of both pools is of greater importance.

Response: We agree that the rate of change is important, which is why we talk about C uptake of forest vegetation in the previous sentence. However, we'd also maintain that the size of the pool is important, as it (1) indicates the importance of forest vegetation for the climate system, and (2) illustrates the significance for the climate system if substantial amounts of the stored C are released to the atmosphere through novel disturbances. After careful consideration we have thus decided to retain the statement in its original form.

Line 36; "...has removed many dispersal barriers for species"

Response: Changed as suggested.

Lines 47-55: The selection of studied species is interesting, because of the comparison/contrast between very different taxa. Do the authors consider these to be the most active/aggressive pests and pathogens? Why these species were selected could be justified further by a rewording of a sentence or two.

Response: We agree that it is important for the reader to have more detailed information on how the five pest species were selected. We have thus extended the description of the methods section in this regard, and now explicitly state the four criteria that were used to choose the five study organisms (see lines 252-261).

Lines 86-93. When the authors consider risk to forest C in this section of the Results, they are not considering the fate of the detrital C (or of salvage logging). This forest C doesn't disappear, so the way this section is written seems to overinflate the impact of the 5 species, as per the major comment above.

Response: Please see our detailed comments above. We in fact report two indicators of C impact, one which does not consider the fate of the detrital C (live C at risk), and one that does (equilibrium C cycle effect). This has been clarified throughout the text by revising and extending the description of our methodology.

Line 97. The wording "theoretical model of C dynamics" is odd. The underlying algorithms for the MODIS NPP calculations are derived from Biom BGC, so it might be better to say "process-based model" or "mechanistic model"

Response: Here we actually do not refer to the model that is behind the remotely sensed NPP but to the model that we used to derive recovery times, which is a theoretical model of disturbance impacts on the carbon cycle developed by Weng and coworkers (Weng et al., 2012, J. Geophys. Res. 117, G03014). In order to prevent any confusion for the reader here we have (i) added the reference to the statement and (ii) improved and extended our description of the methods to more clearly describe our approach.

The approach the authors use to calculate recovery times is effective. However, in this analysis, do the authors consider stand age? This has been a more difficult factor to tease out

from the MODIS and remote sensing data, because predicted NPP is mostly driven by estimated leaf area from NDVI and some climate data. This seems relevant to the average time to recovery that is presented in Figure 2, because for example Mediterranean stands are younger on average than temperate stands.

Response: Stand age is indeed not explicitly considered in our calculations here. The proxy used in our model is C residence time (parameter τ_1 in Eq. 1), which contains considerable information regarding the spatial variation in stand ages across Europe. As expected by the Reviewer, a map of τ_1 (which was now added to the Supplementary Material as Supplementary Figure 13) shows considerably shorter residence times in the Mediterranean system, and is thus also contributing to higher C recovery rates in Mediterranean forests. This fact was now added to the section where we discuss differences in recovery time between Mediterranean and boreal systems (lines 124-126). We thank the Reviewer for the suggestion to look into this!

Also, Figure 2 is essentially a “worst case scenario” and it would be helpful to remind the reader that this is not only “all areas suitable” for invasion but also represents some range in the percentage of tree mortality in the legend.

Response: We agree that it is important to also note the mortality rates that underlie this figure. We have thus added information about the mortality rates to the figure caption as suggested.

Lines 136-137. This almost seems like too strong of a statement, and also ignores many of the other services that European forests have provided. Perhaps reword to encompass a broader scope?

Response: We agree that forest ecosystems in Europe fulfill a wide range of ecosystem services. But as the scope of our article is on C storage and climate regulation we find it

proper to begin the discussion section with a statement on the importance of forest C. Nonetheless, we agree that the alien pest species considered here could also have strong detrimental impacts on other ecosystem services. These issues are broad to the reader's attention in the final paragraph of the discussion section, which was improved and extended in the revised version of the manuscript (lines 222-240).

Lines 146-154. This is where the potential impact of higher temperatures and pathogens and predators of the five invasive species could be mentioned. It is set of uncertainties that could affect their impact to forest carbon storage and productivity.

Response: We agree with the Reviewer on the importance of highlighting SDM uncertainties, and have thus thoroughly revised and extended this part of the text. Specifically, this section is now three times longer than in the original submission, and addresses potential issues resulting from correlative (and not process-oriented) modeling, emerging biotic interactions, and data limitations in species distribution modeling.

Lines 155-156 and 176-177. This paragraph would benefit from a bit more detail about the fate of C during disturbance. If the authors actually consider "forest C storage" as they state, then they have to include C in dead trees and detritus derived from these contrasting disturbances. If they did this, the times to recovery presented in Figure 2 would be much shorter, because then NPP needs only to catch up to the rate of CWD decomposition.

Response: Please see our detailed responses above. We here report on both indicators that do and do not consider the fate of C during disturbance (live C at risk and equilibrium C cycle effect, respectively). We have now considerably revised and extended the description of our methodology and have added a number of figures and analyses to the Supplement to make our approach more clear for the reader.

Lines 170-173 and Table 1. The authors largely mention reactive management strategies to these pest and pathogens, but some proactive forest management treatments such as thinning and prescribed burning have shown to be effective at reducing the impact of some invasive species (although not the five that authors consider). The authors should consider mentioning this, especially for ALB and PWN.

Response: We agree with the Reviewer and have added also examples for proactive management treatments in the revised version of the text (lines 233-235).

Table 1. The ranges of mortality for ALB and PWN seem quite high. Does this include any management activities? “Quarantining infested areas” is true, but if the authors could provide more specific treatments, that would help inform forest managers. The addition of “not selling infected wood or moving firewood” or other more specific management practices would be helpful where appropriate.

Response: The mortality range reported here is indeed for unmanaged infestations. This information was added to the Table caption. Table S3 in the Supplement contrasts these to the (considerably lower) rates typically observed when infestations are addressed by pest management. In general, Table 1 provides an overview for the reader on the most important management measures for each forest pest species. “Quarantining infested areas” is an overarching measure that inter alia includes a ban on exporting any potentially infested wood (products) from infested regions, and thus also includes the activity that has been highlighted by the reviewer. For more detailed information on the specifics of management measures, the references cited in Table S3 provide the reader with relevant further information.

Most of the methods are sound, and the authors should be commended for the integration of many sources of information in this synthesis. It would be interesting to follow up on some of

the species distribution models (both pests and trees) to detail where any high temperature effects occur, as mentioned above.

Response: Thank you for the encouraging words on our multi-methods approach – this is also where we see the strength and novelty of our contribution. As for the high temperature limitations in the SDMs please see our response to a similar previous comment.

Carbon stocks of Europe's forests: The methods the authors use to estimate C in live tree biomass is sound, and field data from the forest census plots are essentially equivalent to USFS Forest Inventory and Analysis inventory data collected in the United States. Do these inventory data also include dead trees and CWD on the forest floor? This is typically sampled less frequently, but if the authors were to estimate true forest C storage, these data should be included, as per comments above.

Response: Please see our responses to similar comments made above. CWD and the forest floor was considered implicitly in our model through the differential residence times of total ecosystem C (including CWD and litter C) and live tree C (i.e., the portion that is affected directly by disturbance). The mathematical derivation and proof can be found in Weng et al. (2012, J. Geophys. Res. 117, G03014). We have revised the description of the methods to better explain our approach for the reader.

Lines 353-365. How accurate were the attributed tree species groups to the actual species composition in the field inventory data? This is likely beyond the scope of this manuscript, but it does have some bearing on the estimate of mortality, especially for pines vs. other conifers.

Response: To further elucidate the question of how accurate our wall-to-wall C estimates for the host tree species of the pests were we conducted an additional analysis based on the national forest inventory data of Austria, France, Italy, Norway, Spain and Sweden. For these

countries, we calculated the live C in the host tree species of the pests from forest inventory data, and compared the upscaled estimate to the one derived from our wall-to-wall C maps (Supplementary Fig. 8). This analysis showed that at the level of country and pest species, our remotely sensed estimates corresponded well with the terrestrial estimate from FIA data. In a regression analysis between the two data sources the intercept was not significantly different from zero (intercept= 1.468, SE= ± 13.953) while the slope was not significantly different from unity (slope= 1.037, SE= ± 0.109). Graphical analysis showed the biggest deviation for Beech Bleeding Cancer in France, where the C in host tree species was considerable overestimated by our approach based on remote sensing (see Fig. R2). However, the overall correspondence was good, with a coefficient of determination between the two data sources of 0.763.

Fig. R2: Comparison of carbon in host tree species estimated from terrestrial forest inventory data (x-axis) and using the remote sensing procedure applied in this study to derive spatially explicit C maps (Supplementary Fig. 8). Each point denotes a combination of country (Austria, France, Italy, Norway, Spain and Sweden) and host tree species group.

The methods used to estimate live biomass accumulation (NPP) using MODIS data seems sound. The simple equation used to calculate C recovery time is adequate. Where do the estimated residence times of C in live biomass come from? This is where it would be useful to know two parameters; the spatially averaged age of forests in the simulation unit, and the maximum age and biomass of each tree species in each simulation unit. Because many of these forests are intermediate in age, they likely have not come close to reaching maximum age or biomass.

Response: We thank the Reviewer for his/ her assessment of the soundness of our approach. Live C residence times were calculated by dividing live C stocks (Supplementary Figure 7) by annual NPP (Supplementary Figure 9). Both live C stocks and NPP were estimated based on a large data base of FIA plots across Europe (see Neumann et al., 2016, Remote Sensing 8, 554, and Moreno et al. 2017, Geoscientific Data Journal 4, 17-28), consequently the specific forest age structure of Europe's forests is implicitly considered in the residence times derived from these estimates. In order to better illustrate this important intermediate result of our calculations we have added a figure to the Supplementary Material showing the continental scale distribution of live C residence times (Supplementary Figure 13). Furthermore, based on other comments and suggestions we have also added a sensitivity analysis investigating the effect that C residence time estimates have on the C cycle impact of alien pests (see Supplementary Table S7).

Lines 404-410. As the authors state, they do not account for management practices in reducing the distribution of pest species. However, these have shown to be effective in some cases. It seems like this would be a fruitful area for sensitivity analyses for forest mortality, especially for ALB and PWN.

Response: We agree with the Reviewer that this would be an important further line of inquiry. However, since we did not explicitly account for the spread of the pests, testing different

assumptions about management options limiting their spread cannot be studied within our current analysis framework.

Lines 433-439. This is a key point and should be elevated to the Discussion section.

Response: We agree that the distinction between C at risk and equilibrium C cycle effects (accounting for the residence times of C in the system, the system recovery, as well as the fate of the live C that is killed by the pests) are important. Here, we assess both of these values. We have considerably revised and extended the methods, results, and discussion, in order to make the distinction between these values more clear for the reader.

Reviewer #3

Seidl et al. use species distribution modeling (SDM) techniques to tackle an ambitious, very important problem. That is, providing estimates of potential changes to forest carbon in response to the invasion of pests and pathogens following climate change. Here I have commented on their SDM methods, as requested. The methods are generally well written and likely accessible to a broad audience across multiple disciplines.

Response: Thank you for the positive assessment of our work!

Application of species distribution models to study effects of global environmental change has exploded in popularity over the past decade (See multiple review articles by W. Thuiller, Janet Franklin, and others). A common challenge discussed in this literature is the tradeoff of using simple correlative approaches (as done in this manuscript) for parameterizing static SDMs versus more complex mechanistic approaches that attempt to model dynamic processes of pest population growth and dispersal over larger regions (e.g. Cunniffe et al. 2016 PNAS). The authors have appropriately carried out the simpler correlative approach, which require less data (and thought) on the species' life history strategies. But, they might consider justifying the strength and weaknesses of their approach versus using a dynamic modeling approach. This exercise should also involve stating a number of model assumptions about equilibrium, dispersal etc..., which aren't currently included in the manuscript but are recommended in Thuiller's and Franklin's best practices of SDM applications in environmental change studies.

Response: We agree with Reviewer #3 that a more process-oriented modeling approach, considering the life history traits of the selected species explicitly, would be desirable. However, we are also aware about the considerably increased amount of information that is required for such an approach – information that is in many cases not available for emerging

pest species. As suggested by the Reviewer we have now highlighted this limitation of our SDM approach, and the associated simplifying assumptions made, in the discussion section, and explicitly contrast our correlative approach against a more process-oriented approach. More broadly, we have considerably extended the discussion of limitations of our modeling in order to provide more context for the reader (see also our comments to similar suggestions by Reviewer #2 above).

Seidl et al. only consider first order interactions between pest redistribution and climate change. For example, the possibility of host redistribution and changes to biotic interactions under novel climates are not considered.

Response: We agree that these are major limitations of our approach. Consequently, we have revised and extended the discussion in this regard, to make these limitations clear for the reader (lines 177-197).

Other simplifying assumptions also influence the results. For example, do pests and pathogens really respond to annual precipitation amounts and 50-year climate averages? Probably not, a large literature exists showing how sensitive pests and pathogens are to the timing of specific weather conditions and seasonal variability in host phenology/susceptibility.

Response: We agree with the reviewer on the importance of climatic variability in the context of forest pests and pathogens. While climate averages frame the species' geographical extent of occurrence quite well, more detailed descriptors of the climate at a given site such as timing and sequence of particular weather conditions such as drought, as well as frequency and magnitude of climatic events are likely to also influence the occurrence and timing of specific outbreaks of forest pests. However, using the broadest extent of global occurrence data on our focal species in order to describe their climatic niches to the best degree possible, we frequently lack exact dates of observation, which means that the influence of individual

climatic events cannot be captured in our SDMs. One way forward would be using local or regional scale data collected at fine temporal scales, and link them with broad-scale SDMs in a meta-modelling approach (cf. Talluto et al. 2016, Glob. Ecol. Biogeography 25, 238-249). We agree that this is an important issue and now discuss them explicitly in the main text in lines 160 – 168.

Finally, dispersal rates for future invasion seem to be ignored. The authors should better explain their assumption of complete spread in the future time step.

Response: We thank Reviewer #3 for this suggestion. This limitation of our approach - and how it potentially affects our results - is now discussed in more detail in the revised discussion section (see lines 168-176).

Overall, the methods chosen are sound. There's no single way to implement SDMs in climate change studies. But, the assumptions and their limitations could be more thoroughly described.

Response: Thank you for this overall positive assessment. We agree that it is important to highlight the assumptions and limitations of the chosen modeling approach for the reader, and have revised the text accordingly (see our above responses for details).

Reviewer #4

The paper evaluates the impacts of pest invasion on the carbon storage of European forests. The authors used a niche model to predict the potential distributions of the five major invasive pests for European forests based on current distribution data. And then, they estimated their effects on forest biomass carbon storage using an equilibrium disturbance-carbon storage model. This paper provides very useful baseline evaluation for the risks of forest carbon storage due to pest invasion resulted from climate change. The results are critical for forest management. The paper is well written and the methods are clear and solid.

Response: Thank you for this positive evaluation, and for the helpful comments on how to further improve our work!

For the Equation 1 that is used to calculate the recovery time, my suggestion is to refine the explanation of X_1 and f . This equation describes the recovery time after a pest attack event (i.e., a disturbance event). I put the equation below for convenience:

$$R = -\tau_1 \ln \left(\frac{fX_1 - U\tau_1}{X_{1,0} - U\tau_1} \right)$$

This equation describes the recovery time after a pest attack event (i.e., a disturbance event). Since the authors assume the disturbance event will not change forest productivity (U) and background mortality (reciprocal of τ_1), it's OK to let $U\tau_1$ be the target of recovery. But, according to the model, X_1 is actually a sample of the random variable of forest biomass before the invasion of the alien pests. It's expectation (or mean) is:

$$E[X_1] = U\tau_1 \frac{1}{1 - \sigma_0 \tau_1},$$

where σ_0 , is the disturbance regime before the invasion of the alien pests. That's why the authors can get a reasonable recovery time even with $f=0.99$. So, if the authors let $f=1$ (then

remove it from this equation), they still can get the similar recovery time while keeping this equation clear. Thus, it becomes recovering to the state before pest invasion.

Response: We thank Reviewer #4 for his/ her valuable considerations and extension. The change that Reviewer #4 suggests is indeed elegant, as it addresses both the issue of asymptotic recovery and the disturbance regime before invasion. However, after careful consideration we retained a hypothetical undisturbed state as our reference (as per our initial equations), primarily in order to being able to contrast our results directly to those of the natural disturbance regime. In other words: We wanted to compare the effect of native vs. invasive disturbance agents, rather than derive the effect of invasive disturbance agents *on top of* native disturbance agents. Such a comparison was also requested by Reviewer #1. In order to also make this more clear for the reader we have amended our description in the methods section in this regard (lines 422-424). Nonetheless, we also further investigated the solution suggested by Reviewer #4. In order to get an estimate of how big the differences between our current approach (target value for recovery is fX_I) and the one suggested by the Reviewer

(target value for recovery is $U\tau_1 \frac{1}{1-\sigma_0\tau_1}$) are, we have calculated the theoretical f values that would result from implementing the approach by the Reviewer. Using values for the historical natural disturbance regime in Europe taken from Seidl et al. (2014, Nature Climate Change 4, 806-810) for σ_0 and the equations suggested by the Reviewer, we calculated f at the continental scale. The resulting distribution of f values is shown in Figure R3 below. The results from this analysis illustrate that the f value of 0.99 that we have chosen in our analysis numerically corresponds very well with the approach suggested by Reviewer #4. In other words, both approaches yield similar results with regard to the equilibrium C cycle effect of invasive alien pests.

Fig. R3: f values (recovery fraction relative to equilibrium C carrying capacity) resulting from a consideration of the current natural disturbance regime in Europe. The distribution shows the frequency of values calculated at the level of individual grid cells for the European continent.

Lines 102~103 in Page 4: “Recovery is slowest in boreal forests ... lower annual C uptake than ...”. Here, I think it is because of the high residence time in boreal forests. For the

system $\frac{dX}{dt} = u - \rho X$, the recovery time is a function of turnover rate ρ (i.e., the reverse of tau) because the fraction of recovery is:

$$\frac{X(t)}{u/\rho} = 1 - e^{-\rho t}.$$

Response: The Reviewer is indeed correct that also residence time influences speed of recovery, besides that C input from NPP. Also, in natural boreal forest ecosystems C residence times can indeed be expected to be considerably higher compared to temperate

forests. However, as we here focus only on live tree C, the residence times do not consider the considerable C pool stored in the litter, soil, and peat of boreal forests. Also, the boreal forests of Europe are intensively managed by industrial timber companies, with rotation periods in Fennoscandinavia being in fact below those of Central Europe, where large forested areas are often found in mountain ranges (70-90 years vs. 90-120 years). Consequently, based on aggregated continental-scale forest inventory data, the live C residence times do not differ strongly between boreal and temperate forests in Europe. We have thus refrained from changing the statement as suggested by the Reviewer. We have also added a map with the residence times used to the Supplementary Material (see Supplementary Figure S13).

If the invasive pests stay in these forests forever, rather than a single attack, the disturbance regime shifts from σ_0 , to σ and the equilibrium forest biomass will be:

$$E[X1(\sigma)] = U\tau_1 \frac{1}{1 - \sigma\tau_1}$$

This situation will permanently reduce the carbon storage of European forests. But as for the

disturbance regime after pest invasion ($\sigma = \frac{D}{K+R}$, Equation 3 in Page 19), I cannot figure out

how it is derived. According to the REGIME model, $\sigma = \frac{D}{MRI}$, where D is the mean ratio of the biomass removed by pest attack events and MRI is the mean return time of the events.

How MRI equal to “K+R”? Please explain it.

Response: We agree with Reviewer #4 that our derivation of the mean disturbance interval MRI was not satisfactorily explained in the initial version of the manuscript. We have now revised this part of the text, including the respective equation, to make our rationale more explicit for the reader. Specifically, we assumed that once present, the frequency of an attack by a pest species is limited by the time needed from infection to tree death (K) and by the

ability of the forest to recover the lost tree carbon (R). We thus set $MRI=K+R$. This is now explained more clearly to the reader in the revised text.

Reviewers' comments:

Reviewer #2 (Remarks to the Author):

Invasive alien pests threaten the carbon stored in Europe's forests

Rupert Seidl, Günther Klonner, Werner Rammer, Franz Essl, Adam Moreno, Mathias Neumann, and Stefan Dullinger.

The authors have submitted a very strong revision to their manuscript presenting an analysis of the potential impact of five invasive forest pests and pathogens on live carbon biomass in European Forests, using climate models and information on species ranges and tolerances to project how climate change could affect live tree biomass and subsequent recovery times in forested areas.

Overall, the analyses are sound and the authors use appropriate climate scenarios, recent forest inventory data for baseline tree biomass, and MODIS derived estimates of net primary productivity to derive recovery times. The manuscript is well written and appropriate for Nature Communications.

I feel my comments have been addressed thoroughly, and it was a real pleasure to read through the revision letter. I liked this manuscript when I first reviewed it--my comments were more for clarity than anything else. I feel this manuscript should now be accepted for publication.

Reviewer #3 (Remarks to the Author):

The revised manuscript effectively describes SDM modeling assumptions as well as the trade-offs and rationales for the approaches applied. I'm happy with the revisions to the SDM modeling sections. Best wishes, Ross K. Meentemeyer

Reviewer #4 (Remarks to the Author):

The authors have answered my questions and explained how they obtained equations 1 and 3. I understand how challenging it is to put the diverse data into such a theoretical modeling framework to make projections. The authors have done an excellent job.

People may have different understanding of the theoretical model and always have to make their adaptations to the model in their applications. I don't insist my understanding to the model. Here, I just want to explain the assumptions I made for deriving the analytical solution. Hope they could be of any help for the authors to make their own modifications to the model.

1. Disturbances are Poisson events, meaning their occurrence does not depend on ecosystem states and the disturbances happened before.

The assumption is most criticized by reviewers. We made tests with Weibull distribution that assumes the disturbances are not independent. The results are similar with those from Poisson distribution.

2. NPP and turnover rate are not affected by disturbance events, which gives an analytically solvable recovery pattern.

3. Comparing to the recovery period, the period of disturbance is very short.

So, the authors can take this model as a "null" model and make their own changes for their modeling.

Thanks!

Ensheng Weng

Reviewer #5 (Remarks to the Author):

General comments

This is generally a good paper and quite interesting and can make a valuable and novel contribution.

In brief, I agree entirely with the comments of the original Reviewer #2 when they discuss the confusion around live tree C, which is what was analyzed, and total ecosystem C, which is only included in the equilibrium C calculation in a theoretical and somewhat opaque manner. Note that specific numerical predictions for amounts of C at risk are only given for live tree biomass C. There continues to be confusion around this point, even in this revision, and further clarity around this is needed or the results of this paper will be easily mis-interpreted by less knowledgeable readers, who will be susceptible to presenting these as overestimates of the potential impact on total ecosystem C. You will see in the nature of my specific comments, which I wrote out as I was reading the paper for the first time and mostly did not edit, is that my very first question was around this topic (hmm, how do they account for soil and dead organic matter C, I wonder?).

I remained confused about this point throughout, and it takes until the very end of the methods attached to the main text of the paper to find out how soil and litter C are accounted for, and then if one wants to specifically actually see how, you have to go look up and read the supplement of another paper (Seidl et al. 2014) to find out critical details about how this is actually done (for example, that litter and soil C are combined into a one pool model, which has many known issues when such an approach is used). This is not transparent enough of a description of this key aspect of the analysis, but one that could be easily fixed with careful editing to make this point very clear.

The C recovery calculations are also highly dependent on the predictions of a very generalized theoretical model of forest C dynamics (Weng et al. 2012). I have thoroughly read this paper and my evaluation is that this method is fine for determining very rough general trends (their abstract is correct, when it says “the model allows us to get a sense of the sensitivity of terrestrial carbon dynamics at large scales”). But, one can only get a “sense” of it, the specific numerical values are only rough approximations. In other words, it’s a back of the envelope type calculation (Maybe it’s a fancy envelope, made of organic handmade paper and decorated with artisanal ink calligraphy, but it is still an envelope nonetheless).

While I don’t see this as a severe problem since this analysis is really just about trying to get a sense of the scale of the alien pest risk issue, which is a unique and novel contribution of this paper, it would be helpful if any specific numerical predictions were associated with confidence intervals (you do a very good job of describing qualitatively the uncertainties, which is commendable). Luckily, these are also numerically calculated, at least under some assumptions and for some output indicators (included in the supplement), and it might be of use if these were included in the main text of the paper when results were presented.

Specific comments:

Line 33-36. In this description of disturbances, the way it is written creates confusion between disturbances like wildfire which immediately release C to the atmosphere, and what you are talking about in this paper, which is biotic disturbances (insects and disease) which don't do this, but rather transfer C from living biomass C pools to dead organic matter pools, from which a portion of this is released over time through decay (some is eventually stored for a long time in soil C, buried wood, peat, etc.). As a side note, another effect of biotic disturbance is reductions in productivity (ie trees stay alive, but grow at slower rates). This is not captured, which is fine, but this could be mentioned. Must re-write this section to show how the two types are different.

Line 26, In North America, chestnut blight is probably a better example for alien pests that has nearly eliminated entire species from landscapes (Dutch elm disease can be managed somewhat through sanitation, and elms have not disappeared to the extent that chestnut, *Castanea dentata*, has)

Line 53. Remove "to date"

Line 57 Change "assess" to "estimate"

Line 58 Change "from these novel...." to "of these novel...."

Line 71. Specify very clearly that the C at risk is live biomass C, not total ecosystem C. Do this throughout the paper. I only further call out in the comments some specific examples of

Line 77-78. Again this will cause confusion. Yes, live biomass is recovered through NPP. However, total ecosystem C is recovered through NEP, and needs to account for the decay process of the dead trees.

Line 81-86. Use of the word "potential". It's possible that C storage potential is reduced by alien pests, under the assumption that even if susceptible species are wiped out that other unsusceptible species don't take their place (for example, areas of eastern US formerly covered with American chestnut now host other species). I would reword this section to take out "potential" and make it clear that you are talking about the amount of carbon stored in the live tree pool only, and calculating by how much this is reduced. The potential for the forest to store this C is not affected, or at least, you have not calculated how much its affected (ie forest C "carrying capacity" so to speak is likely unaffected, unless no other species, or less productive species, are capable of growing on lands formerly occupied by the invasive pest susceptible species, in this case potential would actually be reduced).

Line 107-116. The heading says “live tree carbon at risk” but then the text in the paragraph in many cases makes it easy to confuse “live tree carbon” and “total ecosystem carbon”.

Line 226. “marked services”? Do you mean “marketed services”?

Line 235-236. How does one increase tree species diversity, when there are not a lot of species to choose from, or where sometimes only a few or one tree species are really best suited, silviculturally, for growing on a specific site? Do we introduce non-natives? Plant species in unnatural mixes? Or is just planting both spruce and pine, instead of just one or the other, good enough (as an example).

Line 320. Section heading says “Carbon stocks of Europe’s forests”, but the text description in the methods, as far as I can see, talks only about live tree C stock estimation.

Line 359 “fate of the disturbed C”. Based on reading the rest of this section, I don’t see how you do this. In your description of how you applied the Weng et al. 2012 model, you only talk about the recovery parameter τ_1 , which is for live biomass C only. Their model has two other recovery parameters τ_2 , and τ_3 for litter and soil C, and I don’t see this discussed anywhere here.

Line 418. Ah, I see τ_E finally appears here. So one finally now knows that to determine how the litter and soil C are accounted for, one must go and read this previous paper Seidl et al. reference 6. In fact, the details of this are not found until several pages into the supplementary material for that paper (for example, that litter and soil are in fact combined into a single soil and litter C pool)

Line 434-436. Finally this point is made. I would actually move this to much closer to the beginning of the paper. Potentially even into the 1st paragraph of the main text so that there is no confusion.

Reviewer #2

The authors have submitted a very strong revision to their manuscript presenting an analysis of the potential impact of five invasive forest pests and pathogens on live carbon biomass in European Forests, using climate models and information on species ranges and tolerances to project how climate change could affect live tree biomass and subsequent recovery times in forested areas. Overall, the analyses are sound and the authors use appropriate climate scenarios, recent forest inventory data for baseline tree biomass, and MODIS derived estimates of net primary productivity to derive recovery times. The manuscript is well written and appropriate for Nature Communications.

Response: We thank Reviewer #2 for this positive assessment of the revised version of our manuscript!

I feel my comments have been addressed thoroughly, and it was a real pleasure to read through the revision letter. I liked this manuscript when I first reviewed it--my comments were more for clarity than anything else. I feel this manuscript should now be accepted for publication.

Response: Thank you for helping to improve our work further!

Reviewer #3

The revised manuscript effectively describes SDM modeling assumptions as well as the trade-offs and rationales for the approaches applied. I'm happy with the revisions to the SDM modeling sections. Best wishes, Ross K. Meentemeyer

Response: We thank Prof. Meentemeyer for helping to improve the earlier version of our work, and for the kind words on the revised version of our manuscript.

Reviewer #4

The authors have answered my questions and explained how they obtained equations 1 and 3. I understand how challenging it is to put the diverse data into such a theoretical modeling framework to make projections. The authors have done an excellent job.

Response: Thank you!

People may have different understanding of the theoretical model and always have to make their adaptations to the model in their applications. I don't insist my understanding to

model. Here, I just want to explain the assumptions I made for deriving the analytical solution. Hope they could be of any help for the authors to make their own modifications to the model. 1. Disturbances are Poisson events, meaning their occurrence does not depend on ecosystem states and the disturbances happened before. The assumption is most criticized by reviewers. We made tests with Weibull distribution that assumes the disturbances are not independent. The results are similar with those from Poisson distribution. 2. NPP and turnover rate are not affected by disturbance events, which gives an analytically solvable recovery pattern. 3. Comparing to the recovery period, the period of disturbance is very short. So, the authors can take this model as a "null" model and make their own changes for their modeling. Thanks! Ensheng Weng

Response: We are grateful to Prof. Weng for further elaborating on the background of his modeling approach, which we have applied here to assess the impact of novel pest species on the forest carbon cycle. As elaborated here by the originator of the approach, the REGIME model provides a robust and mathematically tractable means to quantify C effects of disturbances in ecosystems. Following Prof. Weng's advice we have indeed made moderate adaptations to the original model in our work in order to tailor it to the study system at hand, e.g. with regard to the effect of forest management and salvage harvesting (aspects that are of particular importance in Europe's forest ecosystems). We agree with Reviewer #5 (see below) that our current quantification provides only an initial estimate of the potential C cycle effects of invasive alien pests that should be refined in the future. However, we also feel that Prof. Weng's REGIME model is a powerful tool for such impact assessments across large spatial scales, as it requires a minimum number of parameters while being logically consistent and mathematically tractable, and thus allowing a robust comparison of the impacts of different forcings (e.g., disturbance agents, climate scenarios, management approaches).

Reviewer #5

This is generally a good paper and quite interesting and can make a valuable and novel contribution.

Response: We thank Reviewer #5 for this generally positive assessment of our work.

In brief, I agree entirely with the comments of the original Reviewer #2 when they discuss the confusion around live tree C, which is what was analyzed, and total ecosystem C, which is only included in the equilibrium C calculation in a theoretical and somewhat opaque manner.

Response: We thank Reviewer #5 for her/ his comments (here and below) regarding a potential confusion of the two response variables addressed in our contribution. We agree that it is crucially important to be clear about the variables assessed, and to avoid any ambiguities and misinterpretations of results by the reader. We have thus thoroughly revised our manuscript in this regard, specifically (i) naming the response variables and their significance (i.e., ecological impact of pests vs. their role in the context of the climate regulating function of forests) explicitly in several instances throughout the text, and (ii) revising the wording to clearly state in each part of the results section which response variable is reported (i.e., consistently rewording from “C at risk” to “live tree C at risk”). For details on the changes implemented see our responses below. In addition, to address the “somewhat opaque manner” of the calculation of equilibrium C cycle effects (step 3 in our analysis) we have improved the description in the methods and materials, adding equations and more details on the specific calculations. We, however, would also like to point out that the entire methodology applied in step 3 of our analysis has been published previously, and we also do not want to insinuate for the reader that this methodology is a novel contribution of the current work.

Note that specific numerical predictions for amounts of C at risk are only given for live tree biomass C.

Response: We respectfully disagree with Reviewer #5 here, and would like to draw the attention of the Reviewer to Table 3 in the main manuscript text (was Table 2 in the previous version of the manuscript), which contains numeric results of our third analysis step, i.e. the assessment of equilibrium C cycle effects. In addition to this display item in the main text, the Supplementary Material presents additional quantitative results on equilibrium C cycle effects, e.g. with regard to the impact of pest management and different climate scenarios (Table S10).

There continues to be confusion around this point, even in this revision, and further clarity around this is needed or the results of this paper will be easily mis-interpreted by less knowledgeable readers, who will be susceptible to presenting these as overestimates of the potential impact on total ecosystem C. You will see in the nature of my specific comments, which I wrote out as I was reading the paper for the first time and mostly did not edit, is that my very first question was around this topic (hmm, how do they account for soil and dead organic matter C, I wonder?). I remained confused about this point throughout, and it takes until the very end of the methods attached to the main text of the paper to find out how soil and litter C are accounted for, and then if one wants to specifically actually see how, you have to go look up and read the supplement of another paper (Seidl et al. 2014) to find out critical details about how this is actually done (for example, that litter and soil C are combined into a one pool model, which has many known issues when such an approach is used). This is not transparent enough of a description of this key aspect of the analysis, but one that could be easily fixed with careful editing to make this point very clear.

Response: We thank Reviewer #5 for suggesting to further improve the clarity of our writing. We have now undertaken the careful editing suggested by the Reviewer, which has resulted in several specifications of the response variable addressed throughout the text (e.g., rewording from “C at risk” to “live tree C at risk”). Also, we now clearly state the two different response

variables addressed here (i.e., live tree C impacts, and equilibrium total ecosystem C impacts) very early in the manuscript (introduction section, lines 88-91), and explicitly mention that they are indicators for different things (i.e., the ecosystems impact of invasive aliens, and their relevance in the context of the climate regulating function of forest ecosystems, respectively). We are confident that these changes alleviate the issue of having to read the entire methods section (which happens to be at the end of the manuscript in Nature Communications) in order to being able to interpret our results. Said methods section was also extended to include more equations and details about our modeling approach (including, for instance, more information on residence times and the C pools explicitly considered in our modeling). Overall, we feel that these careful edits and additions have considerably improved our work further, and we are grateful to Reviewer #5 for suggesting them!

The C recovery calculations are also highly dependent on the predictions of a very generalized theoretical model of forest C dynamics (Weng et al. 2012). I have thoroughly read this paper and my evaluation is that this method is fine for determining very rough general trends (their abstract is correct, when it says “the model allows us to get a sense of the sensitivity of terrestrial carbon dynamics at large scales”). But, one can only get a “sense” of it, the specific numerical values are only rough approximations. In other words, it’s a back of the envelope type calculation (Maybe it’s a fancy envelope, made of organic handmade paper and decorated with artisanal ink calligraphy, but it is still an envelope nonetheless).

Response: We agree that the theoretical model applied here is not able to fully capture the intricacies of the forest carbon cycle. In order to make this more clear to the reader we have added an explicit statement of this limitation to the discussion section (lines 185-190), and have also included a reference on the state-of-the-art and challenges of simulating disturbance impacts on forest carbon (Liu et al., 2011 J. Geophys. Res. 116, G00K08). However, we still feel that the approach chosen here is appropriate and insightful: Considering the high

uncertainties that arise from (i) coarse and inhomogeneous input data availability at the continental scale (e.g., C stored in forest soils), and (ii) the error propagation from the model linking performed here (SDM to C impact model, cf. Fig. S1), the application of a more complex C impact model would— in all likelihood – have resulted in a considerably lowered signal to noise ratio in our modeling. In other words, the effects of different disturbance agents, management options, and climate scenarios would have been smaller than the uncertainty from applying a complex process-based model, strongly limiting inferential potential. We specifically see the utility and potential of our modeling approach not in assessing absolute values of Tg C lost, but rather in quantifying the relative differences in forcings, for which the simple, tractable REGIME model applied here is well-suited. We have now revised the discussion to make this point more clear for the reader – see lines 185-190 in the revised version of the manuscript.

While I don't see this as a severe problem since this analysis is really just about trying to get a sense of the scale of the alien pest risk issue, which is a unique and novel contribution of this paper, it would be helpful if any specific numerical predictions were associated with confidence intervals (you do a very good job of describing qualitatively the uncertainties, which is commendable). Luckily, these are also numerically calculated, at least under some assumptions and for some output indicators (included in the supplement), and it might be of use if these were included in the main text of the paper when results were presented.

Response: We agree with Reviewer #5 on the importance of uncertainty assessments in general, and on quantifying the effect of different sources of uncertainty on our estimates of C at risk. Consequently, we have extended the estimation of bootstrapped confidence intervals, and have included it in the main text (newly added Table 2). This analysis now explicitly reports on the uncertainties from local variation in C stocks and mortality rates on live tree C

at risk in (lines 122-126). We have also extended our description of the methods to reflect these amendments (see lines 397-407).

Line 33-36. In this description of disturbances, the way it is written creates confusion between disturbances like wildfire which immediately release C to the atmosphere, and what you are talking about in this paper, which is biotic disturbances (insects and disease) which don't do this, but rather transfer C from living biomass C pools to dead organic matter pools, from which a portion of this is released over time through decay (some is eventually stored for a long time in soil C, buried wood, peat, etc.).

Response: We agree with Reviewer #5 that this is an important distinction to make. We have revised this particular section of the introduction, and explicitly mention the potential pathways of C loss resulting from disturbances. Also we have added three new references to the text, underpinning the different pathways of C loss. More generally we'd like to point out that emissions from combustion account for only a relatively small amount of disturbance-related C losses in wildfires (for instance estimated to between 13% and 35% of aboveground C by Meigs et al. 2009, *Ecosystems* 12, 1246-1267). The overall more important effects of disturbances (be it from wildfire, insects or diseases) are increased heterotrophic respiration (due to higher soil temperatures) and decreased C uptake (due to lower leaf area). These issues are now explicitly mentioned in the revised version of the text, increasing the clarity of disturbance-mediated C loss for the reader.

As a side note, another effect of biotic disturbance is reductions in productivity (ie trees stay alive, but grow at slower rates). This is not captured, which is fine, but this could be mentioned. Must re-write this section to show how the two types are different.

Response: We have revised the section to include also the potential for decreased C uptake in response to disturbances. Furthermore, we have added a reference to Peters et al. (2013,

Ecosystems 16, 95-110: “Influence of disturbance on temperate forest productivity”), who discuss this issue in detail.

Line 26, In North America, chestnut blight is probably a better example for alien pests that has nearly eliminated entire species from landscapes (Dutch elm disease can be managed somewhat through sanitation, and elms have not disappeared to the extent that chestnut, *Castanea dentata*, has)

Response: Presumably this statement refers to line 46 (and not 26) in the previous version of the manuscript, which is where we give an example for a detrimental past invasion of an alien pest species. Following the suggestion by Reviewer #5 we have substituted chestnut blight for Dutch elm disease in the text, and have updated the respective reference accordingly.

Line 53. Remove “to date”

Response: Done.

Line 57 Change “assess” to “estimate”

Response: Done.

Line 58 Change “from these novel....” to “of these novel....”

Response: Done.

Line 71. Specify very clearly that the C at risk is live biomass C, not total ecosystem C. Do this throughout the paper. I only further call out in the comments some specific examples of

Response: Reworded to live tree C at risk here and in many other instances throughout the manuscript to avoid confusion.

Line 77-78. Again this will cause confusion. Yes, live biomass is recovered through NPP. However, total ecosystem C is recovered through NEP, and needs to account for the decay process of the dead trees.

Response: As suggested by Reviewer #5 above, the statement here makes explicit that our work focuses on the recovery of live tree C. In the context of live tree C NPP is the relevant measure, as also mentioned by Reviewer #5. While we agree that NEP is another important indicator of recovery we have refrained from adding it to the text here, as we feel that mentioning a variable that is not assessed in the current manuscript would further add to the confusion regarding response variables highlighted by Reviewer #5, rather than resolving it.

Line 81-86. Use of the word “potential”. It’s possible that C storage potential is reduced by alien pests, under the assumption that even if susceptible species are wiped out that other unsusceptible species don’t take their place (for example, areas of eastern US formerly covered with American chestnut now host other species). I would reword this section to take out “potential” and make it clear that you are talking about the amount of carbon stored in the live tree pool only, and calculating by how much this is reduced. The potential for the forest to store this C is not affected, or at least, you have not calculated how much its affected (ie forest C “carrying capacity” so to speak is likely unaffected, unless no other species, or less productive species, are capable of growing on lands formerly occupied by the invasive pest susceptible species, in this case potential would actually be reduced).

Response: We agree that the term “potential” was misleading here. We have now revised the sentence and have omitted the term potential. We also agree on the importance of the replacement of affected species by other species, and discuss this issue explicitly in lines 225-227. Which species could replace affected tree species and to what degree they could compensate C losses remains an important question for research, particularly in the relatively species-poor forests of Europe (compared to Eastern North America or Asia).

Line 107-116. The heading says “live tree carbon at risk” but then the text in the paragraph in many cases makes it easy to confuse “live tree carbon” and “total ecosystem carbon”.

Response: We agree with Reviewer #5 and thank her/ him for this comment. Upon closer inspection of the section the confusion arose from the fact that we used the term “total” when referring to the effect of all five alien pest species taken together. This resulted in ambiguous wording, and some of the statements could have indeed be misinterpreted as meaning total C, i.e., total ecosystem C. We have remedied the situation by explicitly rewording all statements to live tree C, which should clarify our response variable for the reader.

Line 226. “marked services”? Do you mean “marketed services”?

Response: We do indeed – thanks for spotting this mistake!

Line 235-236. How does one increase tree species diversity, when there are not a lot of species to choose from, or where sometimes only a few or one tree species are really best suited, silviculturally, for growing on a specific site? Do we introduce non-natives? Plant species in un-natural mixes? Or is just planting both spruce and pine, instead of just one or the other, good enough (as an example).

Response: This is a very good question – one could basically write a separate paper on this issue! An in-depth analysis of this question is, however, beyond the scope of our current contribution. Nonetheless we agree that our initial statement, which only referred to tree species diversity, was too narrow, as it is not applicable under some ecological conditions (e.g., in high latitude boreal forests or subalpine forests, where the set of potential tree species is strongly limited by harsh environmental conditions). Possible options under such conditions would be to e.g., increase structural diversity rather than species diversity (particularly when pests only affect certain developmental stages of trees), to foster genetic diversity (to increase

the propensity for resistant strains), and to foster landscape heterogeneity (in order to limit spread rates of invasive alien species). We have now revised this section accordingly, and have broadened our statement in order to make it relevant for a wider set of ecological conditions (lines 249-252). Furthermore, we have amended the references cited here to reflect this broader scope of the revised statement.

Line 320. Section heading says “Carbon stocks of Europe’s forests”, but the text description in the methods, as far as I can see, talks only about live tree C stock estimation.

Response: Correct. The section heading was changed to “Live tree carbon stocks of Europe’s forests” in the revised version of the manuscript.

Line 359 “fate of the disturbed C”. Based on reading the rest of this section, I don’t see how you do this. In your description of how you applied the Weng et al. 2012 model, you only talk about the recovery parameter τ_1 , which is for live biomass C only. Their model has two other recovery parameters τ_2 , and τ_3 for litter and soil C, and I don’t see this discussed anywhere here.

Response: The section highlighted by the Reviewer gives an overview over our analysis approach. Our approach is subsequently described in detail in the following paragraphs, which is why the details might not yet be entirely clear at this point (based on the next statement by Reviewer #5 it seems that things become considerably clearer after reading through the entire paragraph). In order to convey the structure of the text more clearly to the reader (overview of the three analyses steps in lines 368-378, followed by a detailed description of these steps in lines 379-465) we have added a sentence explicitly referring to the detailed description following below. In addition, we have clarified our analytical approach and have refined the language with regard to response variables throughout the text, as suggested by Reviewer #5.

Line 418. Ah, I see tauE finally appears here. So one finally now knows that to determine how the litter and soil C are accounted for, one must go and read this previous paper Seidl et al. reference 6. In fact, the details of this are not found until several pages into the supplementary material for that paper (for example, that litter and soil are in fact combined into a single soil and litter C pool)

Response: We'd like to point out that tauE only appears at this point of the methods section (i.e., the description of our third analysis step) as the previous two analytical steps focus on live tree C and thus did not require tauE in their calculations. In order to make this explicit for the reader and avoid any confusion, we have added sentences containing this information in two different sections of the manuscript: First, at the end of the introduction section, in order to alert the reader already early in the manuscript that different response variables are assessed in the different analysis steps (lines 85-89). And second, in the beginning of the methods section pertaining to C impact modeling, in order to once more remind the reader that the methods description that follows relates to different response variables (lines 376-378). Regarding the description of the tauE data used, we'd like to point out that we did apply them as derived in a previous study. Given the constraints of a manuscript in Nature Communications it is not possible to reiterate the full process chain of calculating these values here. Nonetheless, we have extended the methods section in this regard, and have added more information on tauE, including the fact that these were derived jointly for litter and soil pools. We now also explicitly report on the derivation of tauE, and have added an equation to the text detailing how tauS, which is an important constituent of tauE, is calculated (Eq. 3).

Line 434-436. Finally this point is made. I would actually move this to much closer to the beginning of the paper. Potentially even into the 1st paragraph of the main text so that there is no confusion.

Response: We have revised the introduction to early in the text make clear that we here analyze two different response variables, live tree C and total ecosystem C. Specifically, we have added a sentence clearly stating which analyses steps focus on which response variable (lines 88-89). Furthermore, as suggested by Reviewer #5 we have clarified the wording with regard to our response variables throughout the manuscript. Overall, we are confident that our revisions minimize the potential for confusing interpretations of our results.

REVIEWERS' COMMENTS:

Reviewer #5 (Remarks to the Author):

I am satisfied with the responses provided to my review comments and have nothing further to add.